Citation: *Molecular Systems Biology* 9:654
www.molecularsystemsbiology.com

# Plant stem cell maintenance involves direct transcriptional repression of differentiation program

Ram Kishor Yadav[1,2,7], Mariano Perales[1,7], Jérémy Gruel[3,4,7], Carolyn Ohno[5,6], Marcus Heisler[5,6], Thomas Girke[1], Henrik Jönsson[3,4,*] and G Venugopala Reddy[1,*]

[1] Department of Botany and Plant Sciences, Center for Plant Cell Biology (CEPCEB), Institute of Integrative Genome Biology (IIGB), University of California, Riverside, CA, USA, [2] Indian Institute of Science Education and Research, Mohali, India, [3] Computational Biology and Biological Physics Group, Department of Astronomy and Theoretical Physics, Lund University, Lund, Sweden, [4] Sainsbury Laboratory, University of Cambridge, Cambridge, UK, [5] European Molecular Biology Laboratory, Heidelberg, Germany and [6] University of Sydney, Sydney, Australia
[7] These authors contributed equally to this work.
* Corresponding authors. GV Reddy, Department of Botany and Plant Sciences, Center for Plant Cell Biology (CEPCEB), Institute of Integrative Genome Biology (IIGB), University of California, Riverside, CA 92521, USA. Tel.: + 1 951 8273482; Fax: + 1 951 8274437; E-mail: venug@ucr.edu or H Jönsson, Sainsbury Laboratory, University of Cambridge, Bateman Street, Cambridge CB2 1LR, UK. Tel.: + 44 (0)1223 761128; Fax: + 44 (0)1223 350422; E-mail: henrik@thep.lu.se

In animal systems, master regulatory transcription factors (TFs) mediate stem cell maintenance through a direct transcriptional repression of differentiation promoting TFs. Whether similar mechanisms operate in plants is not known. In plants, shoot apical meristems serve as reservoirs of stem cells that provide cells for all above ground organs. WUSCHEL, a homeodomain TF produced in cells of the niche, migrates into adjacent cells where it specifies stem cells. Through high-resolution genomic analysis, we show that WUSCHEL represses a large number of genes that are expressed in differentiating cells including a group of differentiation promoting TFs involved in leaf development. We show that WUS directly binds to the regulatory regions of differentiation promoting TFs; *KANADI1, KANADI2, ASYMMETRICLEAVES2* and *YABBY3* to repress their expression. Predictions from a computational model, supported by live imaging, reveal that WUS-mediated repression prevents premature differentiation of stem cell progenitors, being part of a minimal regulatory network for meristem maintenance. Our work shows that direct transcriptional repression of differentiation promoting TFs is an evolutionarily conserved logic for stem cell regulation.
*Molecular Systems Biology* **9**: 654; published online 2 April 2013; doi:10.1038/msb.2013.8
*Subject Categories:* plant biology; chromatin & transcription
*Keywords:* central zone; CLAVATA3; shoot apical meristem; stem cell niche; WUSCHEL

## Introduction

Genome-wide expression analysis and chromatin immuno-precipitation (ChIP) assays in both mice and human embryonic stem (ES) cells have shown that a core group of transcription factors (TFs) represses several differentiation-promoting TFs by directly binding to their promoters (Boyer *et al*, 2005; Loh *et al*, 2006). Unlike animal systems, stem cell homeostasis in plant systems does not involve cell behaviors such as physical asymmetric cell divisions, oriented cell divisions and cell migration (Reddy, 2008). For example, the shoot apical meristem (SAM) stem cell niche that resides at the tip of each shoot harbours a set of pluripotent stem cells within the central zone (CZ) and provides cells for the development of all aboveground organs (Figure 1A; Steeves and Sussex, 1989). The progeny of stem cells enters the flanking peripheral zone (PZ) where they differentiate into leaves or flowers (Figure 1A). The Rib-meristem (RM)/organizing center (OC) located beneath the CZ functions as niche and provides cues for stem cell specification (Figure 1A; Mayer *et al*, 1998).

It is believed that regulated patterns of cell division rates of stem cells and their progenitors are important in the timely transition of stem cell progenitors to differentiation pathways (Meyerowitz, 1997). In this context, it is essential to understand the molecular logic that underlies stem cell maintenance in the SAM stem cell niche.

The homeodomain TF WUSCHEL (WUS), which is expressed in the OC/niche, has been shown to be necessary and sufficient for stem cell specification in overlying cells of the CZ (Mayer *et al*, 1998; Schoof *et al*, 2000). Besides stem cell specification, WUS restricts its own levels by activating *CLAVATA3 (CLV3)* transcription in the CZ (Figure 1A; Fletcher *et al*, 1999; Brand *et al*, 2000). CLV3, a small secreted peptide, activates CLAVATA1 receptor kinase pathway to restrict *WUS* transcription to few cells, thus forming a feedback system (Clark *et al*, 1997; Ogawa *et al*, 2008). A recent study has shown that WUS protein synthesized in the niche migrates into adjacent cells and activates *CLV3* transcription by binding to the promoter (Yadav *et al*, 2011). This study also reveals that

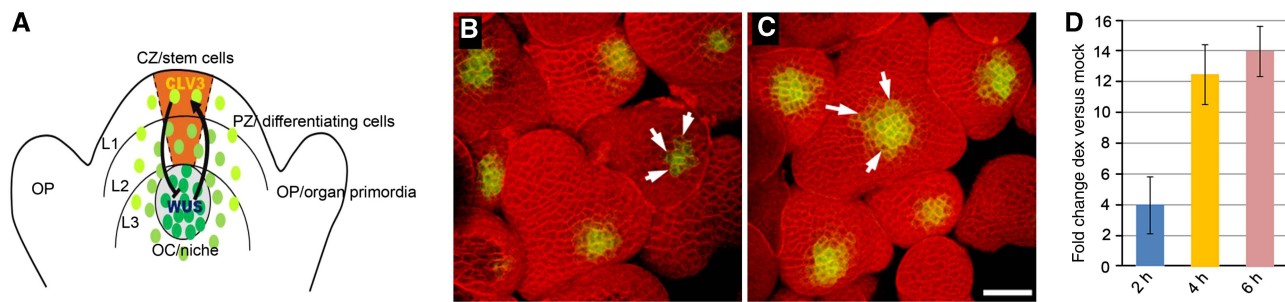

**Figure 1** System for assaying WUS-response genes at higher spatial resolution. (**A**) A schematic of *Arabidopsis* shoot apical meristem (SAM) stem cell niche showing stem cell domain/the central zone (CZ), the organizing center (OC)/niche and differentiating region/the peripheral zone (PZ). Transcriptional domains of *WUS*, *CLV3* and WUS protein gradient are highlighted in different colors. Mutual feedback regulation between *CLV3* and *WUS* is shown. (**B**, **C**) Three-dimensional reconstructed top views of *ap1-1;cal1-1* SAMs carrying Dex-inducible form of WUS (*35S:;WUS-GR*), labeled with stem cell marker-*pCLV3::mGFP-ER* (green) in plants treated with mock and Dex, respectively, are shown. Arrows point to outer limits of stem cell domain and expansion of stem cell domain upon Dex treatment. Scale bar in (C) represents 25 μm and it remains same for (B). (**D**) qRT–PCR analysis showing temporal dynamics of *CLV3* activation upon WUS induction. Error bars represent standard deviation for two biological replicates.

the WUS protein gradient extends into the PZ where stem cell progenitors differentiate (Yadav *et al*, 2011). Understanding of the WUS function in cells of the PZ may reveal molecular logic of stem cell maintenance. Two lines of evidence have suggested that WUS represses differentiation. First, a live imaging study has shown that WUS determines organ primordia size by regulating patterns of differentiation in the PZ (Yadav *et al*, 2010). Second, genomic analysis has revealed that WUS downregulates genes expressed in leaves including genes that encode components of transcriptional repression complex (Busch *et al*, 2010). However, the repressed target genes presented do not provide immediate clues to their function in regulating stem cell homeostasis (Busch *et al*, 2010).

Here, we have developed a WUS-responsive gene network at a higher spatial resolution, which reveals that WUS represses several key TFs implicated in various aspects of leaf differentiation. By employing biochemical methods, we further show that WUS directly binds to the promoter regions of *KANADI1*, *KANADI2*, *YABBY3* and *ASYMMETRIC LEAVES2*, TF encoding genes that are expressed in the PZ and mediate early aspects of leaf differentiation. We develop a cell-based computational model of the SAM by integrating WUS-mediated direct transcriptional repression of differentiation with a CLAVATA/WUSCHEL regulatory network model. The differential equation model predicts that WUS function is required to prevent premature differentiation of stem cell progenitors. We validate model perturbation predictions through transient manipulation of WUS levels followed by live imaging, which shows that WUS-mediated direct transcriptional repression is required to keep differentiation program at a distance from the meristem center/stem cell domain.

## Results

### Development of a high spatial resolution WUS-responsive gene network

To explain the function of WUS in stem cell maintenance, we developed a high-resolution spatial map of the WUS-regulated transcriptome. An earlier study has employed FACS-mediated purification of distinct cell types of the stem cell niche to develop a cell type-specific high-resolution map (Yadav *et al*, 2009). The method that is based on isolating fluorescently labeled cell populations from SAMs of *apetala1-1;cauliflower1-1* (*ap1-1;cal1-1*) double mutants has been shown to accurately report gene expression patterns and also to be highly sensitive in detecting transcripts of ~1000 genes that have not been detected, in any developmental stages, when whole tissues were used (Yadav *et al*, 2009). We introduced a dexamethasone (Dex)-inducible form of *WUS* consisting of the WUS protein-coding region fused to the ligand-binding domain of the rat glucocorticoid receptor (GR), expressed under a ubiquitous promoter (*35S::WUS-GR*) into *apetala1-1;cauliflower1-1* (*ap1-1;cal1-1*) double mutant background. *35S::WUS-GR;ap1-1;cal1-1* SAMs that were treated with 10 μM Dex solution for 12 h revealed an increase in stem cell domain as revealed by *CLV3* (a WUS target gene) promoter activity when compared with mock-treated SAMs (Figure 1B and C; Yadav *et al*, 2011). WUS-mediated activation of *CLV3* was observed as early as 2 hs after Dex application with *CLV3* expression reaching maximum levels by 4 h and saturating by 6 h after Dex application (Figure 1D). Therefore, SAMs were treated with 10 μM Dex solution for 4 h and RNA samples were hybridized to *Arabidopsis* ATH1 gene Chip (Affymetrix). To identify putative genes that are directly regulated by WUS, SAMs were, in independent experiments, simultaneously treated with 10 μM Dex and 10 μM Cycloheximide (Cyc), protein synthesis inhibitor and Cyc alone for 4 h. A comparison of Dex-treated samples with mock identified 641 genes as differentially expressed (DEGs) ($\geqslant$/$\leqslant$2-fold; $P < 0.01$), which consisted of 238 upregulated genes and 403 downregulated genes (Supplementary Tables 1 and 2). While a comparison of Dex + Cyc with Cyc identified 457 genes as DEGs ($\geqslant$/$\leqslant$2-fold; $P < 0.01$), which consisted of 154 upregulated genes and 303 downregulated genes (Supplementary Tables 1 and 3). Next, we compared the overlap between genes that respond to WUS upon Dex treatment alone with Dex + Cyc treatment by constructing a four-way Venn diagram (Supplementary Figure 1). This analysis revealed a set of 49 upregulated genes and 140 downregulated genes that were common to both treatment conditions (Supplementary Figure 1).

## WUS represses a group of differentiation promoting TFs

To determine the role of WUS-responsive genes, we mapped their patterns of expression on the various sub-domains of the SAM. An earlier study has shown that 2361 genes are differentially expressed ($\geqslant$2-fold; $P>0.01$) across three cell types including the CZ, the RM and the PZ (Yadav *et al*, 2009). To visualize the spatial distribution of WUS-responsive genes, we superimposed DEGs on to the cell type-specific gene expression map. In all, 17% (40 genes out of 238) transcripts activated by WUS in Dex versus mock comparison ($\geqslant$2-fold; $P>0.01$) were mapped to the CZ and the RM while only 5% transcripts (11 genes) were found in the PZ cells (Figure 2A; Supplementary Table 1). An opposite trend was observed for genes that were downregulated wherein 38% (152 genes out of 403) ($\leqslant$2-fold difference; $P>0.01$) transcripts were either mapped to the PZ or their expression was detected in both the PZ and the RM. Only 10% (42 genes) were found in the CZ and the RM (Figure 2B; see Supplementary Table 1), showing that WUS represses a large group of genes expressed in differentiating cells. The genes that were not mapped to any of the three cell types may be broadly expressed in SAMs. Alternately, the resolution of the expression map that involves just three cell types may be limited to resolve genes enriched in individual cell layers of the SAM since *CLV3* expression domain overlaps with that of WUS (Yadav *et al*, 2009). In Dex + Cyc versus Cyc comparison, 17% (26 genes out of 154) WUS-activated genes mapped to the CZ and the RM while only 8% (12 genes) were part of the PZ cells (Figure 2C; Supplementary Table 1). Among 303 downregulated genes, only 11% (33 genes) were mapped to the CZ and the RM while 29% (87 genes) of them were either part of the PZ or their expression was detected in both the PZ and the RM (Figure 2D; Supplementary Table 1). Activation of *CLV3* and repression of *ARABIDOPSIS RESPONSE REGULATOR 7* and *15* were observed in the presence of Cyc (Supplementary Table 1), which is consistent with them being direct transcriptional targets of

WUS (Leibfried *et al*, 2005; Zhao *et al*, 2010; Yadav *et al*, 2011). To exclude artifacts, if any, from *ap1-1;cal1-1* SAMs, we re-analyzed a select set of genes identified in the microarray experiments by performing qRT–PCR analysis on RNA samples extracted from finely dissected wild-type SAMs expressing *35S::WUS:GR*. This analysis showed that all six of the activated targets tested including *CLV3* were activated within 4 h of WUS induction in both the absence and the presence of Cyc (Figure 2E and F). A similar agreement between microarray analysis and qRT–PCR analysis was observed for 14 of the repressed targets analyzed in both the absence and the presence of Cyc (Figure 2G and H). Taken together, these results reveal that a majority of WUS-activated genes are broadly expressed within the central parts of SAMs and that WUS represses a large group of genes expressed in the PZ. A similar trend was observed in Cyc-treated SAMs, suggesting that WUS may repress genes expressed in differentiating cells of the PZ through a direct transcriptional control.

Among genes downregulated in the presence of Cyc, 37 were annotated as TFs (Supplementary Table 1). The repressed TFs included key TFs implicated in leaf polarity establishment and differentiation such as *KANADI 1* and *2* (*KAN1* and *KAN2*) (Kerstetter *et al*, 2001), *ASYMMETRIC LEAVES 2* (*AS2*) (Lin *et al*, 2003) and *YABBY 3* (*YAB3*) (Kumaran *et al*, 2002; Figure 2G and H). Genes encoding homeodomain TFs, *KNAT1/BREVIPEDICELLUS (BP)* (Figure 2G and H) and *BLH5 (BELL1-LIKE HOMEODOMAIN 5)*, that have been shown to be expressed in the PZ and the RM of SAMs, and implicated in cell fate specification and inflorescence stem growth were also repressed by WUS (Lincoln *et al*, 1994; Douglas *et al*, 2002; Venglat *et al*, 2002; Bhatt *et al*, 2004; Rutjens *et al*, 2009). This apart, WUS represses *BHLH093*, a FAMA class of basic helix-loop-helix TF required to promote differentiation of stomatal guard cells of leaves (Ohashi-Ito and Bergmann, 2006) *ANAC083/VND-INTERACTING 2* (Yamaguchi *et al*, 2010), C2C2 domain-DOF Zinc Finger TF-*DOF2.4* and *AT4G24060* implicated in vascular development in leaves (Gardiner *et al*, 2010). The repressed

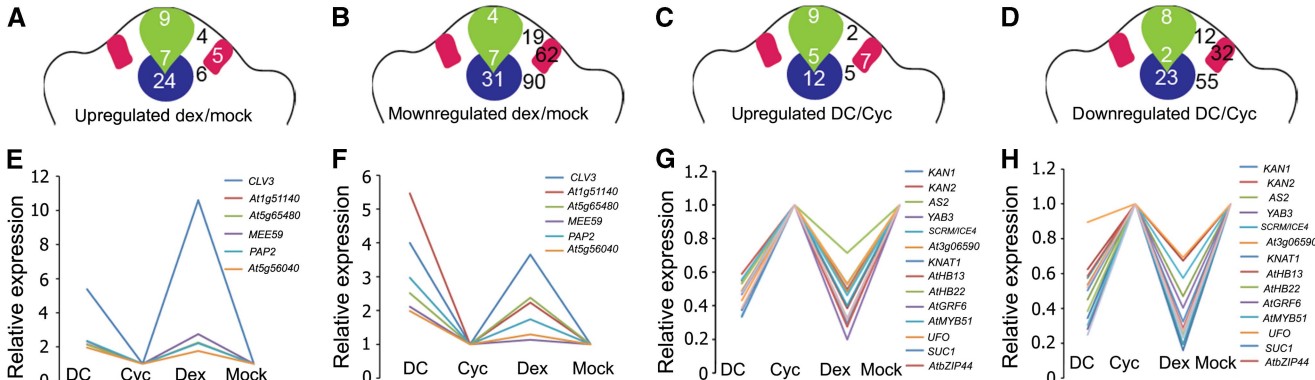

**Figure 2** A spatial map of WUS-responsive genes. (**A**, **C**) Genes upregulated by Dexamethasone (Dex)-inducible WUS in the absence and presence of Cycloheximide (Cyc), respectively. Genes downregulated by WUS in the absence (**B**) and presence (**D**) of Cyc. Numbers on the CZ, the PZ and the RM refer to the number of WUS-responsive genes that are expressed in each domain. Numbers indicated in between cell types are the WUS-responsive genes that are expressed in two adjacent cell types. Normalized gCRMA values (using multiexperiment viewer (MEV4) software) showing relative expression levels of representative set of genes activated (**E**) and repressed (**G**) by WUS in various treatments shown on x axis. Expression profiles of WUS-regulated genes shown in (E) and (G) were confirmed by real-time qRT–PCR, relative expression levels for various treatments indicated on x axis are shown for activated (**F**) and repressed (**H**) genes. Profile plots for few genes in (E) are not visible as these genes share similar expression profiles as others.

genes also included *SCARECROW (SCR)* that encodes GRAS-domain TF and *GROWTH-REGULATING FACTOR 6 (GRF6)*, both are involved in leaf growth (Kim *et al*, 2003; Dhondt *et al*, 2010). We analyzed the RNA localization patterns of a select set of differentiation promoting TFs repressed by WUS (Supplementary Figure 2). Expression of *KAN1, KAN2* and *AS2* was restricted to few cells of the PZ of both SAMs (Supplementary Figure 2A–C) and floral meristems (Supplementary Figure 2D–F). Taken together, our results show that WUS represses a group of differentiation promoting TFs thus excluding them from stem cells of the CZ.

## WUS represses differentiation promoting TFs by binding to their regulatory regions

Next, we tested whether WUS directly binds to the regulatory regions of repressed TFs by performing ChIP coupled to qPCR analysis (ChIP-qPCR) by using anti-WUS (peptide)-antibodies described in an earlier study (Yadav *et al*, 2011). We obtained immunoprecipitated DNA that was enriched in specific genomic regions of *KAN1, KAN2, AS2* and *YAB3* genes (Figure 3A–D). We used electrophoretic mobility shift assays (EMSAs) for fine mapping of WUS binding sites within these genomic regions by testing a series of short oligonucleotides for their ability to bind to WUS (Figure 3E). DNA sequences of *KAN1, KAN2, AS2* and *YAB3* that bound WUS revealed the presence of conserved TAAT core sequences (Figure 3F). Previous studies have shown a similar WUS binding specificity to TAAT containing sequence elements in EMSAs

(Lohmann *et al*, 2001; Leibfried *et al*, 2005; Busch *et al*, 2010; Yadav *et al*, 2011). In some cases, for example the sequence of KAN2 650, AS2 − 480, AS2 750 and YAB3 500 oligonucleotides contained more than one TAAT motif (Figure 4H). Therefore, we explored the WUS binding specificity of *KAN1, KAN2, AS2* and *YAB3* promoter-specific oligonucleotides by introducing mutations into all TAAT motifs present in them (Figure 4H). Mutations within the TAAT core sequences abolished WUS binding, suggesting that WUS binding to these sequences is specific (Figure 4A–G). In cases where more than one TAAT core was detected, a selective lack of binding was observed with mutations in certain TAAT elements and not others (Figure 4H). This analysis shows that TAAT sequences are important but not sufficient to determine WUS binding specificity. In addition, a 'super shift' was observed when *KAN1* promoter oligonucleotides (Figure 3G) were incubated with anti-WUS antibody confirming the presence of WUS protein in WUS-DNA complex. We next tested whether the WUS-binding element found in the *KAN1* promoter can modulate transcription in a heterologous transient expression system by examining the LUCIFERASE (LUC) reporter expression levels in protoplasts isolated from mesophyll cells of *Arabidopsis* leaves. These results show WUS-dependent repression of *LUC* when *KAN1* promoter context containing the WUS-binding element was used (Figure 4I). In contrast, the WUS-dependent repression of *LUC* was abolished when *KAN1* promoter context carrying a mutated WUS-binding element was used (Figure 4I). Taken together, these results demonstrate that WUS protein binds to the regulatory regions of differentiation promoting TFs to repress their expression.

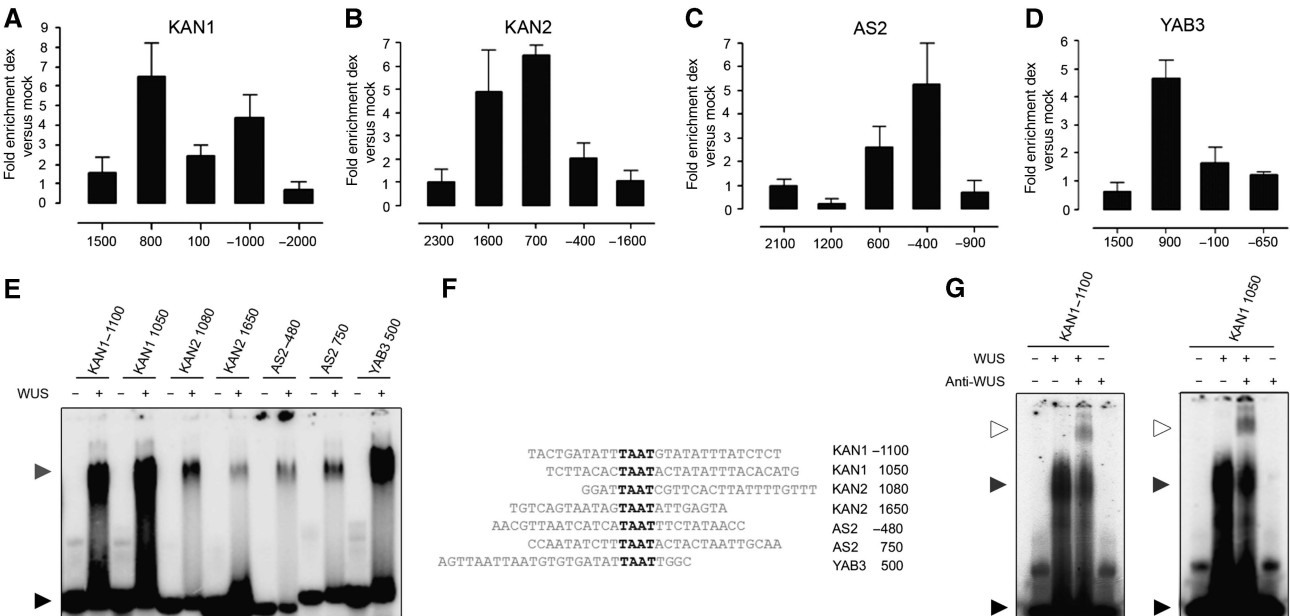

**Figure 3** Differentiation promoting transcription factors are direct transcriptional targets of WUS. (**A–D**) Chip-qPCR showing relative enrichment of regulatory regions of *KAN1, KAN2, AS2* and *YAB3*, respectively. Error bars represent standard deviation. Genomic regions are mapped with respect to transcription start site ( + 1). (**E**) EMSA showing recombinant WUS protein bound to radiolabeled-oligonucleotides corresponding to *KAN1, KAN2, AS2* and *YAB3* regulatory regions. A black arrowhead indicates free probe and a dark gray arrowhead band shift for *KAN1, KAN2, AS2* and *YAB3* radiolabeled oligonucleotides. respectively. The sequences of WUS bound oligonucleotides are shown in (**F**) and conserved TAAT elements are highlighted. (**G**) EMSA showing recombinant WUS protein bound to radiolabeled *KAN1* − 1100 and *KAN1* 1050 oligonucleotides and in the absence and presence of anti-WUS antibody. A black arrowhead indicates free probe, a dark gray arrowhead indicates band shift for *KAN1* − 1100 and *KAN1* 1050 and a white arrowhead indicates 'super shift'.

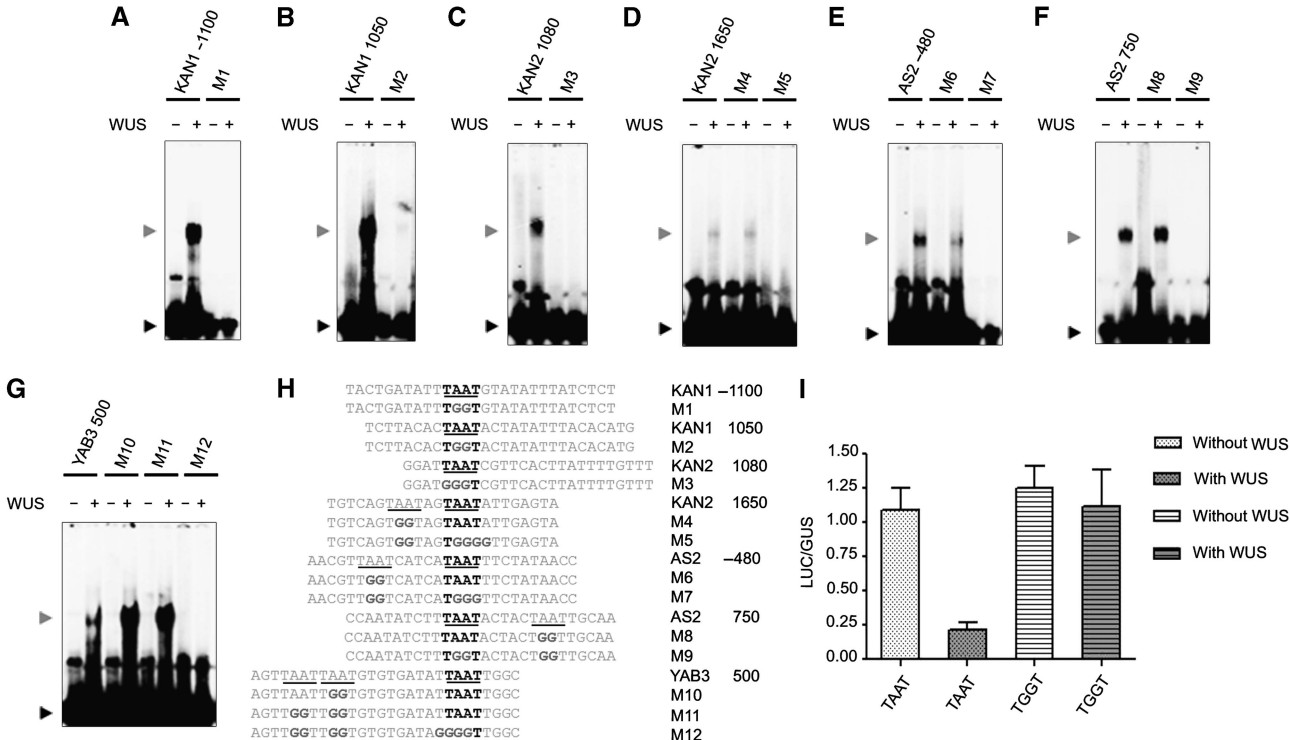

**Figure 4** TAAT elements are required for WUS binding and repression. (**A–G**) EMSAs showing recombinant WUS protein bound to radiolabeled oligonucleotides corresponding to *KAN1*, *KAN2*, *AS2* and *YAB3* regulatory regions and mutant oligonucleotides versions of conserved TAAT sequences. A black arrowhead indicates free probe and a dark gray arrowhead band shift for *KAN1*, *KAN2*, *AS2* and *YAB3* radiolabeled oligonucleotides, respectively. The sequences of WUS bound oligonucleotides and mutants are shown in (**H**). All TAAT elements are underlined and TAAT elements that are essential for WUS binding are shown in bold letters. (**I**) The TAAT promoter element is essential for WUS-dependent repression of *KAN1* in *Arabidopsis* leaf mesophyll protoplasts. Transient transfection assay plots showing repression of LUCIFERASE (LUC) reporter when cloned downstream of a region containing WUS-binding element found in *KAN1* promoter (KAN1 + 950, + 1150:35s::LUC) and a mutated version (TGGT) of *KAN1* promoter. The constructs were tested for transactivation of LUC by cotransfection with or without WUS. The cotransfection of *UBIQUITIN::GUS* served as an internal control. Activity was expressed as a ratio of firefly LUC/GUS activity. Three biological replicates were used for each experiment and the error bars represent the standard deviation.

## A computational model reveals the importance of direct transcriptional repression of differentiation in stem cell homeostasis

Thus far, our analysis suggests that stem cell maintenance may partly be a result of collective repression of differentiation promoting TFs. To understand the importance of the direct transcriptional repression of differentiation genes by WUS, we implemented these interactions in a differential equations model of a three-dimensional SAM tissue, and integrated it with a model of the core CLV3-WUS negative feedback stem cell regulatory network (Figure 5A; Supplementary information; Yadav *et al*, 2011). The model includes the direct activation of *CLV3* transcription from the diffusing WUS protein (Yadav *et al*, 2011), as well as activation from a signal originating from the epidermal cell layer (Jönsson *et al*, 2003; Yadav *et al*, 2011). Although a signal originating in the L1 is yet to be identified in molecular experiments, we propose such a signal based on the behavior of a meristem model in specific mutants in which *CLV3* expression is limited to the outermost cells even when the SAM undergoes dramatic morphogenetic changes (e.g., Brand *et al*, 2002; Supplementary Figure 3I; Supplementary information). The expression of *WUS* is negatively regulated from a diffusive CLV3 signal

(Fletcher *et al*, 1999; Brand *et al*, 2000), and is activated by a static cytokinin/AHK4 signal located at the center of the OC (Supplementary Figure 4; Jönsson *et al*, 2005; Gordon *et al*, 2009; Yadav *et al*, 2011). In this model, we introduced interactions between WUS and the PZ, where *KAN1* was used as a PZ representative. *KAN1* expression as revealed by *pKAN1::KAN1:GFP* (*KAN1* promoter driving the expression of *KAN1:GFP* translational fusion) shows that it is expressed in the outer edges of the PZ (Figure 5E). We introduced a direct transcriptional repression of *KAN1* by the WUS protein as revealed by this study. To generate a sharp *KAN1* expression close to the epidermis, we also introduced an activating signal originating from the L1 layer (Figure 5A). Note that this addition only keeps the *KAN1* expression close to the epidermis, all results reported in this study would be the same without it, except that the PZ gene would be expressed also in internal layers of the PZ. An earlier study has shown that ubiquitous misexpression of *KAN1* results in SAM termination which was argued to be a consequence of abaxialization of leaves (Kerstetter *et al*, 2001). We tested whether ubiquitous *KAN1* misexpression leads to downregulation of *WUS* levels by using Dex-inducible KAN1 (*35S::KAN1:GR*). An RT–PCR analysis of *35S::KAN1:GR* seedlings exposed to 10 µM Dex for 5 days revealed a downregulation of *WUS* transcript levels

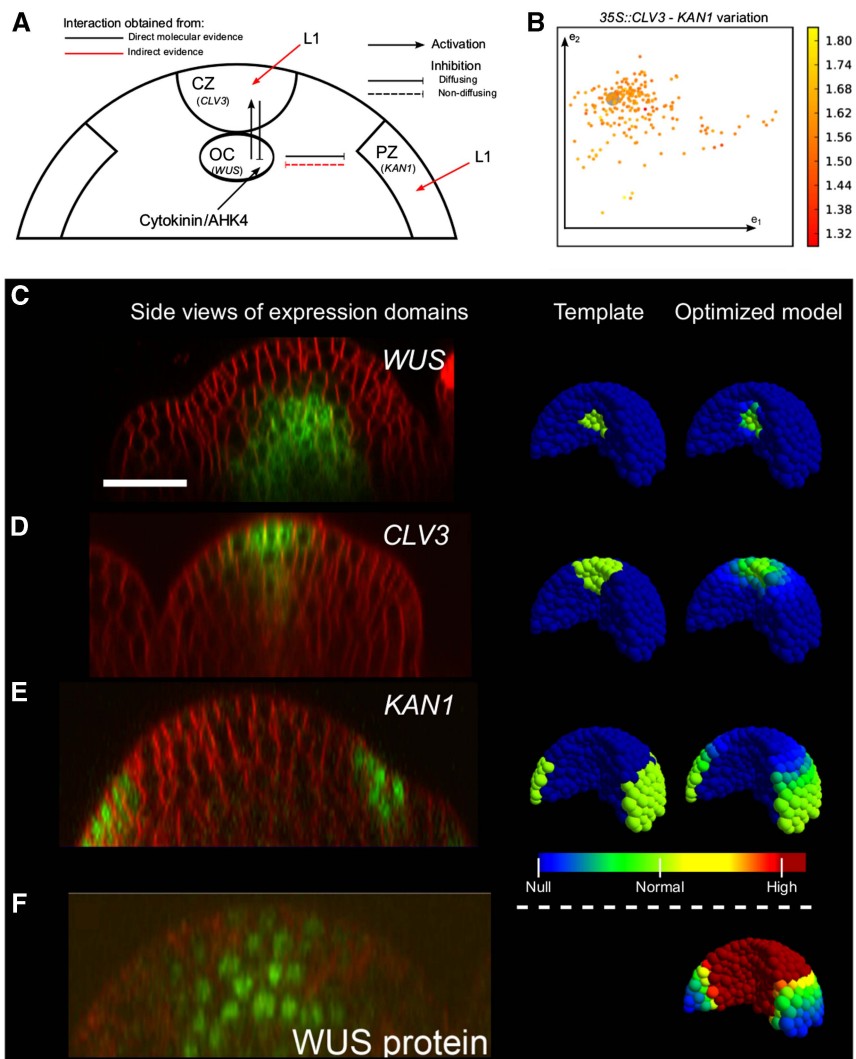

**Figure 5** Design and optimization of the computational model. (**A**) Illustration of the transcriptional interactions in the model mapped on typical functional domains/cell types of SAMs. (**B**) The 229 parameter sets from the optimization procedure, displayed using principal component analysis (first and second principal components: $e_1$, $e_2$). The color scale represents the variation of equilibrium *KAN1* expression between the wild type and a perturbation (transient ubiquitous *CLV3* expression). The gray disk is centered at the parameter set used in all spatial example simulations (Supplementary Table 8). In (**C–F**), first column shows Confocal side views of SAMs showing cell boundaries (red) and RNA or protein distribution domains (green). Corresponding gene names are given on each panel. Scale bar shown in (C) represents 25 μm (same for C–F). Second column: Templates used for optimization showing corresponding gene expression patterns (green). Third column: Simulation output from the example optimized parameter set. The color scale label 'normal' indicates the template defined gene expression level. (C–E) Expression patterns, templates and optimized model for *WUS* (taken from Yadav *et al*, 2009), *CLV3* (taken from Reddy and Meyerowitz, 2005) and *KAN1*, respectively. (F) Distribution of *WUS* protein in the SAM (taken from Yadav *et al*, 2011) and in the model. The color scale for WUS concentration in the model is capped at the concentration value of WUS repressing *KAN1* to half its maximal expression ($k_{w/K}$).

(Supplementary Figure 5). An earlier study has also shown that *YABBY3* and *FILAMENTOUS FLOWER (FIL)* which are expressed in differentiating cells of organ primordia are required to restrict *WUS* and *CLV3* expression domains through a non-cell autonomous mechanism (Goldshmidt *et al*, 2008). We identified *YAB3* as one of the genes directly repressed by *WUS* (Figures 2C, 3D and E). From these data, we use KAN1 as a representative PZ signal modeled as a non-diffusing protein repressing *WUS* expression. Since this interaction has not been verified at the molecular level, we used the model to investigate how different interactions from the PZ on the CLV3/WUS network behaved when ubiquitous

expression of *KAN1* was introduced (Supplementary Figure 6; Supplementary information). The only way in which long-term downregulation of *WUS* could be achieved was if KAN1, possibly indirectly, represses WUS or alternatively, the WUS activation network (cytokinin or AHK4 in our model, Supplementary information). Transcription is modeled with Hill functions and proteins are linearly created based on RNA concentrations (Materials and methods; Supplementary information). RNA and proteins undergo mass action degradation, and CLV3 and WUS diffuse between neighboring cells. An optimization strategy was implemented to obtain sets of parameter values able to fit the expression domains of

the three-dimensional tissue template (Supplementary information). The domains were defined using confocal microscopy data (Figure 5C–E). Using this strategy, 229 parameter sets were extracted in which the model successfully describes the wild-type expression domains (Figure 5B–E) and the WUS gradient (Figure 5F), showing that the regulatory network with a core where WUS acts as a hub with negative feedback with the CZ and positive (doubly negative) feedback with the PZ is sufficient to organize SAMs.

We also analyzed models where individual regulatory interactions were removed. None of the reduced models were able to match experiments, suggesting that the model represents a minimal set of interactions for this network (Supplementary information).

## Transient gene manipulations and live imaging support repression model

We next tested the effects of transient manipulation of WUS levels on the stem cell niche in live imaging experiments. To transiently increase WUS level, we introduced Dex-inducible *35S::WUS:GR* into plants carrying both *pKAN1::KAN1:GFP* (nuclear localized) and the stem cell marker-*pCLV3::mGFP5-ER* (endoplasmic reticulum localized GFP). Within 24 h of Dex application, a dramatic expansion of the *pCLV3* expression was observed that was accompanied by the loss of *pKAN1* expression ($n=4$) (Figure 6A and B). The corresponding perturbation was applied to the computational model

(Supplementary information). Consistent with the experimental observations, the model predicts a large expansion of the *CLV3* expression domain along with a reduction in *KAN1* expression (Figure 6E and F), which was a robust result for all 229 parameter sets optimized for wild-type behavior (Table I; Supplementary Figure 7E). Conversely, the downregulation of WUS levels, achieved through the Dex-inducible overexpression of *CLV3* (*35S::GR:LhG4;6XOP::CLV3*) (Supplementary Figure 8), resulted in a gradual misexpression of *KAN1* in stem cell progenitors that are located closer to the stem cell domain revealing premature differentiation of stem cell progenitors ($n=6$) (Figure 6C and D; Supplementary Figure 9). Introducing ubiquitous *CLV3* expression in the model (Figure 6G; Supplementary information) resulted in a dramatic reduction in *WUS* expression (Supplementary Figures 7F and 3E), and in a gradual shift of *KAN1* expression toward the CZ (Figure 6H; Supplementary Movie 1). The *KAN1* expansion was, as opposed to the experimental result, continued throughout the CZ (Figure 6D and H), suggesting additional *KAN1* repressors in the CZ, although a similar result could also be achieved in the model with a relatively weaker *CLV3* overexpression (weaker 35S promoter activity) (Supplementary Figure 10). Earlier studies have shown that ubiquitous misexpression of *KAN1* (Kerstetter *et al*, 2001) and *AS2* (Lin *et al*, 2003) leads to differentiation and termination of SAM growth. We observed SAM and floral meristem termination upon misexpression of *KAN1* under the influence of the *WUS* promoter (Supplementary Figure 11), which when applied in the model led to disappearance of

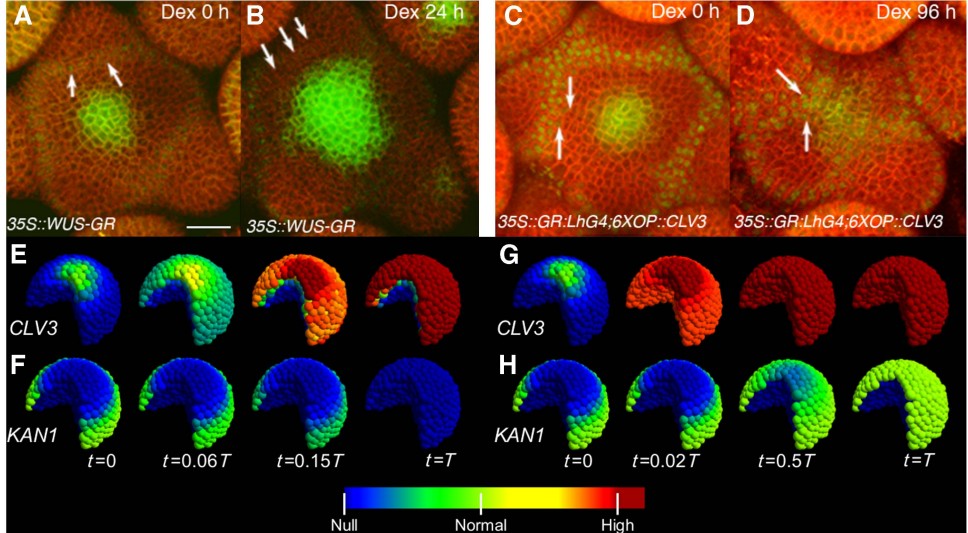

**Figure 6** Dynamics of reorganization of stem cells and differentiating progenitors upon transient manipulation of WUS levels. (**A–D**) 3D-reconstructed top views of SAMs labeled with stem cell/CZ -*pCLV3::mGFP5-ER* (endoplasmic reticulum localized GFP) and the *pKANADI1::KANADI1:GFP* (nuclear localized GFP) expressed in differentiating cells located at the outer edges of the PZ. Plasma membrane localized YFP highlights outlines of all cells (red). Scale bar shown in (A) represents 25 μm and it remains same for (A–D). Arrows in all panels point to KAN1 expressing cells. (A, B) A time-lapse series showing *pCLV3* and *KAN1* expression before (A) and 24 h after (B) Dex-mediated overexpression of *WUS* (*35S::WUS:GR*). Note expansion of *pCLV3* along with loss of *pKAN1* expression (B). (C, D) A time-lapse series showing *pCLV3* and *pKAN1* expression prior to (**C**) and 96 h after (D) Dex-mediated overexpression of *CLV3* (*35S::GR:LhG4; 6XOP::CLV3*), a complete time-lapse series is given in Supplementary Figure 9. Note a progressive shift of *KAN1* expression toward the receding central zone (D). Time-course evolution of modeled *CLV3* (**E, G**) and *KAN1* (**F, H**) expression upon transient ubiquitous expression of *WUS* (E, F) or *CLV3* (G, H). Note that (G) shows both native and induced *CLV3* expression, and the native promoter activity is decreasing due to the loss of WUS. Time points (*t*) are fractions of the time from perturbation induction to model stabilization (*T*). The color scale label 'normal' indicates the template defined gene expression level. Upon *WUS* overexpression, a substantial increase in *CLV3* and decrease in *KAN1* expression domains are observed in both experiment and simulation. Upon *CLV3* overexpression, *KAN1* expression domain extends toward the CZ in both experiment and simulation. Additional model perturbations are displayed in Supplementary Figure 3.

**Table I** Model perturbations

| Perturbation | WUS | CLV3 | KAN1 | Phenotype | References |
|---|---|---|---|---|---|
| *Loss-of-function mutants—Supplementary Figures 7A, B, 3B, and C* | | | | | |
| WUS | ↑ | ↓ (s) | ↓ | Meristem arrest | Brand *et al* (2002) |
| CLV3 | ↑ (s) | ↑ (s) | ↓ | Enlarged meristem | Reddy and Meyerowitz (2005) |
| KAN1 | -(p) | -(p) | -(p) | No effect | Kerstetter *et al* (2001) |
| *Ubiquitous expression mutants—Supplementary Figures 7E–G and 3D–F* | | | | | |
| WUS | ↑ | ↑ (s) | ↓ (s) | Flat meristem | Yadav *et al* (2010) and Figure 6 |
| CLV3 | ↓ (s) | ↑ (s,q) | ↑ (s) | – | Brand *et al* (2000), Reddy and Meyerowitz (2005) and Figure 6 |
| KAN1 | ↓ (q) | ↓ | ↑ | Proposed to arrest SAM | Kerstetter *et al* (2001) and Supplementary Figure 5 |
| *Overexpression mutants—Supplementary Figures 7C, D, 3M, and N* | | | | | |
| WUS | ↓ | ↑ | ↑ | – | – |
| CLV3 | ↓ (q) | ↓ | ↑ | – | Brand *et al* (2000) and Muller *et al* (2006) |
| KAN1 | – | – | – | – | – |
| *Misexpression mutants—Supplementary Figures 12A–F and 3G–L* | | | | | |
| pWUS::CLV3 | ↓ | ↑ or ↓ | ↑ | – | – |
| pWUS::KAN1 | ↓ (p) | ↓ (p) | ↑ | Meristem arrest | Supplementary Figure 11 |
| pCLV3::WUS | ↑ (s) | ↑ (s) | ↓ | Enlarged meristem | Brand *et al* (2002) |
| pCLV3::KAN1 | ↓ | ↓ | ↑ | – | – |
| pKAN1::WUS | ↑ or ↓ | ↑ | ↓ | – | – |
| pKAN1::CLV3 | ↓ | ↑ | ↓ | – | – |
| *Transport rate mutants—Supplementary Figures 12G,H and 3O* | | | | | |
| WUS | ↑ | ↓ (p) | ↑ | Meristem arrest | Yadav *et al* (2011) |
| CLV3 | ↑ | ↑ or ↓ | ↓ | – | – |

Summary of model behavior in simulations using all 229 parameter sets extracted via model optimization. The WUS, CLV3 and KAN1 columns indicate if a perturbation leads to an increase (↑), decrease (↓) or no change (–) of the expression. Red signs indicate cases with published experimental observations. (s) indicates spatial data and (p) phenotypic observation and (q) quantitative mRNA measures. Details of the implementations of the perturbations are given in Supplementary information, the change for individual parameter sets is given in Supplementary Figures 7 and 12, and the spatial changes are displayed in Supplementary Figure 3.

*WUS* and *CLV3* expression (Supplementary Figures 12B and 3H). Taken together, these observations show that WUS-mediated transcriptional repression of differentiation promoting factors prevents premature differentiation of stem cell progenitors.

## The integrated computational model provides a systems description of gene expression regulation in a growing and proliferating SAM tissue

To further connect the new regulatory (repression) interactions from this study with previous work, a total of 17 perturbations including loss-of-function and ectopic expression of *CLV3*, *WUS* and *KAN1*, as well as transport malfunction of CLV-WUS network components were applied to the computational model (Table I; Supplementary information). Simulations with the 229 different parameter sets show some variance in the quantitative behavior (Supplementary Figures 7 and 12), but interestingly the model interaction network is robust in the qualitative (increase versus decrease) response to almost all perturbations (Table I). All perturbations correspond qualitatively with experimentally measured changes in expression domains, where available, and other simulations can be seen as predictions of changes. Loss-of-function mutants have been repeatedly studied in the literature and these mutants were implemented in the model at the protein level making it possible to investigate the changes in mRNA levels of the perturbed gene due to the network interactions.

The *clv3* loss-of-function shows an increase in *WUS* transcripts and an increase in *CLV3* promoter activity and subsequent expansion, and these simulations are consistent with experimental observations made upon transiently downregulating *CLV3* expression (Reddy and Meyerowitz, 2005), although our static template cannot address the increase in SAM size. The PZ region is predicted to move outwards in these simulations (Supplementary Figure 3C). For the *wus* loss-of-function, *CLV3* expression decreases (Brand *et al*, 2002), and in the model this is at least a four-fold decrease (Supplementary Figure 7A). *KAN1* expression is predicted to move inwards, comparable to the ubiquitous *CLV3* expression simulations (Figure 6H), but since it is not a null-mutant implementation (the amount of WUS is strongly decreased, but not set to zero), *KAN1* expression does not cover all of the CZ (Supplementary Figure 3B). Ubiquitous CLV3 and WUS expression simulations are described previously (Figure 6), and simulations with ubiquitous *KAN1* expression lead to extremely low *CLV3* and *WUS* expression (Supplementary Figures 7G and 3F), in agreement with the decrease in *WUS* levels (Supplementary Figure 5), and the terminating SAM phenotype (Kerstetter *et al*, 2001). The same is true for the *pWUS::KAN1* simulations (Supplementary Figure 3H, confer Supplementary Figure 11), and the *pCLV3::WUS* simulations show the non-trivial change of both *CLV3* and *WUS* expressions into high levels in the outermost three layer of cells (Supplementary Figure 3I), similar as reported in a previous model (Yadav *et al*, 2011), and as seen in experiments (Brand *et al*, 2002). *KAN1* expression is predicted to be lost in the meristem in these simulations.

Detailed descriptions of all other mutant simulations are given in Supplementary information.

The model is robust to local parameter perturbations (Supplementary Figure 13), and from the perturbations it is possible to follow some mechanisms in the model. The perturbation of parameters for KAN1 levels do not influence CLV3 or WUS at all; KAN1 does not influence the core network results including when perturbed, except for when KAN1 is expressed ectopically. More important is the dependence of CLV3/WUS on each other, where the direct interactions are seen, for example, when gene expression activities are altered ($V_W$, $V_C$), while the feedback mechanism leading to homeostasis is also indicated, for example, an increased CLV3 (WUS) production ($P_C$, $P_W$) leads to decreased *CLV3* (*WUS*) expression. When the model was challenged in simulations including growth and proliferation (Supplementary Movie 2), the three studied expression domains showed little variation of size and expression intensity as cells moved through the different meristem regions. Taken together, the model shows that the WUS-mediated regulatory network links the CZ and the PZ, and provides sufficient mechanistic framework to explain stem cell homeostasis in a proliferating SAM tissue.

### Other WUS-responsive gene functions and pathways

Besides activation of *CLV3* and repression of differentiation promoting TFs, performing enrichment analyses of gene ontology (GO) terms (Supplementary Table 4) assessed other molecular pathways enriched in the WUS-regulated transcriptome in the presence of cycloheximide (Supplementary Table 3). The WUS upregulated transcriptome contained genes involved in developmental processes (GO:0032502; $P < 0.05$). This included *ANAC018* and *CUP SHAPED COTYLEDON1* (Takada *et al*, 2001) and *EXCESS MICROSPOROCYTES1* (*EMS1*), a receptor-like kinase that promotes cytokinesis during anther development (Zhao *et al*, 2002). *CUC1* that encodes an NAC-domain containing TF has been shown to express in presumptive cells that give rise to SAMs in early embryonic development (Takada *et al*, 2001). Therefore, WUS activation of *CUC1* expression may be relevant in establishment or maintenance of embryonic SAMs. ARR4, a type-A response regulator implicated in negatively regulating cytokinin signaling (To and Kieber, 2008) and seven genes related to response to plant hormone abscisic acid (GO:0009737; $P < 0.01$) were also activated by WUS. WUS also activates *GLABRA1* implicated in epidermal cell fate specification (GO:0001708; $P < 0.01$). Taken together, WUS activates genes implicated in diverse plant processes including those that are critical for stem cell maintenance. The WUS downregulated transcriptome contained significantly enriched GO categories that represent indole derivatives and biosynthesis processes (GO:0042435, $P < 8.19 \times 10^{-5}$), regulation of gene expression (GO:0010468, $P < 0.002$) and cell-to-cell communication and the catalyzation of post-translation modification (GO:0043687, $P < 0.01$) (Supplementary Table 4). Auxin signaling has been shown to promote columella distal stem cell differentiation in *Arabidopsis* root (Ding and Friml, 2010)

and promote leaf differentiation in SAMs (Veronoux *et al*, 2010). WUS represses *ANTHRANILATE SYNTHASE ALPHA SUBUNIT1* (*ASA1*) that encodes a rate-limiting step in tryptophan biosynthesis, *TRYPTOPHAN AMINOTRANSFERASE RELATED2* (*TAR2*) that encodes a protein similar to *TRYPTOPHAN AMINOTRANSFERASE OF ARABIDOPSIS1* involved in IAA biosynthesis and *PHYTOALEXIN DEFICIENT3* (*PAD3*) that encodes a cytochrome P450 enzyme that catalyzes the conversion of dihydrocamalexic acid to indole derivative camalexin (Mano and Nemoto, 2012). The finding that WUS inhibits auxin biosynthetic genes may add another layer of regulation in preventing differentiation of stem cells in addition to the repression of differentiation promoting TFs. However, a high-resolution functional analysis of these genes is necessary to integrate them into the current model of stem cell maintenance.

## Discussion

WUS has been shown to activate transcription of its own negative regulator *CLV3* by directly binding to the promoter regions (Yadav *et al*, 2011). Here, we have shown that WUS represses transcription of differentiation promoting TFs by directly binding to the promoter regions. Thus, the WUS-mediated repression of differentiation program observed in this study links the PZ with the well-established feedback loop between cells of the CZ and the RM through a direct transcriptional control. The dual function of WUS in regulating its own level by activating a negative regulator and repressing differentiation would lead to a robust homeostatic mechanism by balancing stem cell numbers and differentiation rates of stem cell progenitors in a dynamic cellular environment. Our computational model, integrating the WUS-PZ interactions extracted in this work with WUS-CZ interactions, highlights this by robustly generating the spatial patterns for *CLV3* and *KAN1* expression, and the use of the multiple (17) model perturbations connects our findings with more than a decade of previous experimental work. Since no interaction in the model can be removed without losing ability to explain some experimental work, it also suggests that the current model provides a minimal model for regulating stem cell maintenance in the SAM. WUS has been shown to form a concentration gradient with higher levels in the OC/RM and lower levels in adjacent cells (Yadav *et al*, 2011). It is intriguing how a single WUS gradient results in the transcriptional activation of some genes, and the repression of other genes in placing a negative regulator at a distance from its own domain of expression while keeping the differentiation program repressed in stem cell progenitors, and the computational model resolves this by having different WUS thresholds for CZ and PZ regulation. WUS has been shown to act both as an activator and as a repressor of transcription (Ikeda *et al*, 2009). WUS utilizes DNA binding cis-elements containing similar core sequences (TAAT) both to activate *CLV3* transcription (Yadav *et al*, 2011) and to repress expression of differentiation promoting TFs (Figure 3F). Therefore, WUS may utilize spatially localized co-activators and co-repressors to modulate transcriptional output. The computational model requires an L1-derived signal to position *CLV3* expression

domain within the CZ (Figure 5A; Jönsson *et al*, 2003; Yadav *et al*, 2011), which is consistent with the requirement of additional factors that may function along with WUS. One such spatial signal could be the local synthesis of active cytokinins in superficial cell layers as suggested by the expression patterns of *LONELYGUY* class of genes (Yadav *et al*, 2009; Chickarmane *et al*, 2012). Genome-wide expression profiles that have been described will form a rich resource in identifying regional factors that may modulate WUS function (Yadav *et al*, 2009). WUS binds to some TAAT containing sequences but not all (Figure 4A–H), suggesting that sequence context may determine binding specificity. However, a comparison of WUS binding sequences of several WUS-regulated genes did not reveal any consensus sequences around the TAAT core (Figures 3F and 4H). Shape analysis of cis-element sequences that bind *Drosophila* homeodomain TF-HOX has revealed that the width of the minor grove determines the binding specificities (Rohs *et al*, 2009; Slattery *et al*, 2011). Future studies unraveling genome-wide WUS binding patterns along with sequence and shape analysis of WUS binding sites may provide insights into WUS binding specificity and transcriptional modulation.

The use of multiple parameter value sets for the model, and the large consistency for these to respond to perturbations shows the robustness of the modeled SAM gene regulatory network architecture, also during growth. Interestingly, the model predicts that the WUS and CLV3 domains can respond differently to perturbations, indicating the possibility for variability within plant populations. Our methodology, using populations of models (parameter values), is a step toward integrating statistical analysis of experimental population behavior within a modeling framework, something that might be necessary for elucidating the exact regulatory interactions present in the SAM.

Our work shows that WUS functions to prevent premature differentiation of stem cell progenitors. The computational model shows that the experimentally extracted interactions where WUS binds directly and represses the expression of TFs involved in PZ differentiation are sufficient to explain the dynamic changes caused by transient perturbations where expression domains grow or shrink at the boundaries. However, transient depletion of WUS fails to induce differentiation of long-term stem cells within the CZ as suggested by lack of misexpression of auxin-sensitive differentiation marker shown in an earlier study (Yadav *et al*, 2010) and lack of misexpression of *KAN1* (Figure 6D; Supplementary Figure 9J). Leaf formation from the CZ was seldom observed in *wus* mutants instead they continue to produce leaves albeit aberrantly from a defective SAM in a 'stop and go' fashion (Laux *et al*, 1996). This suggests that long-term stem cells may be active in these mutants and argues for function of overlapping mechanisms in protecting long-term stem cells from differentiation signals, as also suggested by the computational model showing a stronger expression phenotype compared with the experiment. The WUSCHEL-LIKE HOMEOBOX5 (WOX5) TF of the WUS family is expressed in the quiescent center of the root meristem and has been shown to repress differentiation of neighboring initials/stem cells in a non-cell autonomous manner

(Sarkar *et al*, 2007). This suggests that WOX-mediated suppression of differentiation of stem cell progenitors is a conserved theme in both shoot and root stem cell niches. Future work should reveal whether WOX5 shares mechanistic similarities with WUS with regard to the nature of non-cell autonomous signaling and molecular mechanisms of repression of differentiation.

An increase in WUS levels leads to de-differentiation of stem cell progenitors into stem cells (Reddy and Meyerowitz, 2005; Yadav *et al*, 2010). Our work provides a molecular basis for de-differentiation wherein WUS-mediated direct repression of differentiation program plays an important role. Deciphering how stem cell progenitors are maintained in WUS-responsive state requires an understanding of the molecular control of the cellular memory system. Studies on murine ES cells have identified bi-valent stretches of both H3 lysine 27 methylation, a gene repressive chromatin mark and H3 lysine 4 methylation, a gene activating chromatin mark within genes which encode differentiation promoting TFs that are repressed by master regulatory stem cell promoting TFs (Bernstein *et al*, 2006; Mikkelsen *et al*, 2007). This suggests that the bi-valent domains are responsible for keeping differentiation genes silent in ES cells, while keeping them primed for activation. Profiling of chromatin modifications in individual cell types of SAM stem cell niche, by using methods described earlier (Yadav *et al*, 2009), should reveal whether collaboration between repression of differentiation and epigenetic pathways is a conserved theme in maintaining stem cell progenitors in a flexible state.

# Materials and methods

## Plant growth and live imaging

Plant growth, live imaging and phenotypic analysis were performed as described earlier (Yadav *et al*, 2010; Yadav *et al*, 2011).

## Plasmid constructs, selection of transgenic lines and rescue analysis

The details of the *35S::WUS:GR* (Yadav *et al*, 2010) and *pCLV3::mGFP5-ER* (Reddy and Meyerowitz, 2005) transgenic lines have been described. The *pKAN1::KAN1:GFP* is a translational fusion that includes 4.95 kb of upstream sequence plus the entire coding sequence with introns fused as a translational fusion to eGFP. *35S::KAN1:GR* and *35S::GR-LhG4;6XOP::CLV3* plants have been described in Wu *et al* (2008) and Yadav *et al* (2010) respectively.

## WUS induction and microarray experiments

Four-week-old *35S::WUS-GR;ap1-1;cal1-1* plants were either treated with 10 μM solution of Dex or in combination with 10 μM cyclohex-imide (Cyc) for 4 h. Mock-treated or Cyc-treated plants served as controls. RNA was isolated from SAMs of 30–35 plants for each treatment by using RNeasy kit (Qiagen). Probe synthesis and array hybridizations were carried out as described earlier (Yadav *et al*, 2009). Microarray data analysis was performed in R using BioConductor packages. The probe set to gene locus mapping for ATH1 gene chip, removal of redundant probe sets and present call analysis was carried out as described earlier (Yadav *et al*, 2009). To assess differentially expressed genes, raw data CEL files were normalized using GCRMA package in R (Supplementary Table 5). DEG was determined with LIMMA package using the normalized expression values (Supplementary Table 1). The Benjamini and Hochberg method was

selected to adjust *P*-values (*P*) for multiple testing and to determine FDR. As confidence threshold an adjusted $P \leqslant 0.01$ was used (Yadav *et al*, 2009). A gene was determined as enriched in a cell type when it was two-fold upregulated. A similar method was used to determine WUS-response genes. A gene was considered as a DEG if it was two-fold upregulated or downregulated ($P < 0.01$) in pairwise comparisons of Dex versus mock or Dex + Cyc versus Cyc. The MAS5 normalized expression values for three cell types were used to generate present (P), marginal (M) and absent (A) calls (Supplementary Table 6). The present call information (PMA values) from the Wilcoxon signed rank test of the MAS5 algorithm is provided for three cell types in Supplementary Table 6. The probe set showing present calls in all three replicates of a sample was considered as positive. GO analysis was performed as described in an earlier study (Yadav *et al*, 2009).

## qRT–PCR

Four-week-old *35S::WUS-GR* wild type SAMs were either treated with 10 µM solution of Dex or in combination with 10 µM cycloheximide (Cyc) for 4 h. Mock-treated or Cyc-treated plants served as controls. *35S::KAN1:GR* plants were exposed to Dex treatment for 5 days before RNA isolation. RNA was isolated using RNeasy kit (Qiagen). cDNA was reverse transcribed using ThermoScript RT (Invitrogen). qRT–PCRs were performed either using sensiMix SYBER kit (Bioline) or SYBR green (BIO-RAD) on a BIO-RAD iQ5 Cycler. Three replicates were used for each reaction and *UBIQUTIN1* was used to normalize the mRNA levels.

## ChIP and qPCR analysis

ChIP and qPCR were performed on 4-week-old *35S::WUS:GR; ap1-1;cal1-1* inflorescences, upon Dex or Mock treatment for 24 h, as described earlier (Yadav *et al*, 2011). Previously published anti-WUS (peptide) antibodies were used (Yadav *et al*, 2011). qPCR analysis by using primers overlapping the gene body and regulatory regions of *KAN1 KAN2, YAB3* and *AS2* (Supplementary Table 7 for list of primers). Three independent sets of biological samples were analyzed.

## Electrophoretic mobility shift assay

His-WUS fusion protein was expressed in BL21 *Escherichia coli* after induction with 1 mM of IPTG for 5 h at 30°C and purified as described earlier (Yadav *et al*, 2011). Single-stranded complementary oligonucleotide fragments corresponding to regulatory regions of *KAN1 KAN2, YAB3* and *AS2* (Supplementary Table 7) were radiolabeled as described earlier (Yadav *et al*, 2011). DNA-protein binding reaction and electrophoresis was performed as described earlier (Yadav *et al*, 2011).

## Protoplast isolation and transactivation assay

Mesophyll protoplast isolation and transactivation was performed as described earlier (Yadav *et al*, 2011) by using 200 bp sequence containing WUS binding element in *KAN1* $-1050$ region.

## Computational modeling and algorithms

The model is simulated on a meristem tissue template comprising 1366 (spherical) cells defined by spatial coordinates and size. Cell neighborhood is defined from the overlap between the spherical cells. The expression domains of *WUS*, *CLV3* and *KAN1* are manually defined with confocal microscopy data used as a template. The differential equation model describes gene expression regulation using Hill functions. Production of peptides and proteins, along with degradation are modeled with mass action kinetics, and transport between cells is passive (diffusion like). Two positional cues are included (*WUS* activator and epidermal cell layer; Supplementary Figure 4) producing three diffusing gene activators. The dynamics of

the molecular concentrations are described by a system of differential equations given by

$$\frac{d[C]}{dt} = V_C \frac{[a1c]^n}{k_{a1c/C}^n + [a1c]^n} \frac{[w]^n}{k_{w/C}^n + [w]^n} - g_C[C]$$

$$\frac{d[c]}{dt} = P_c[C] - g_c[c] + D_c\Delta[c]$$

$$\frac{d[W]}{dt} = V_W \frac{[a2]^n}{k_{a2/W}^n + [a2]^n} \frac{k_{c/W}^n}{k_{c/W}^n + [c]^n} \frac{k_{k/W}^n}{k_{k/W}^n + [k]^n} - g_W[W]$$

$$\frac{d[w]}{dt} = P_w[W] - g_w[w] + D_w\Delta[w]$$

$$\frac{d[K]}{dt} = V_K \frac{[a1k]^n}{k_{a1k/K}^n + [a1k]^n} \frac{k_{w/K}^n}{k_{w/K}^n + [w]^n} - g_K[K]$$

$$\frac{d[k]}{dt} = P_k[K] - g_k[k]$$

$$\frac{d[a1c]}{dt} = P_{a1c}[A1] - g_{a1c}[a1c] + D_{a1c}\Delta[a1c]$$

$$\frac{d[a1k]}{dt} = P_{a1k}[A1] - g_{a1c}[a1k] + D_{a1k}\Delta[a1k]$$

$$\frac{d[a2]}{dt} = P_{a2}[A2] - g_{a2}[a2] + D_{a2}\Delta[a2]$$

where $[X]$ represents concentration of molecule $X$, upper (lower) case represents mRNA (protein) concentrations. The network is present in each individual cell (cell indices have been omitted) and $C$, $W$ and $K$ are CLV3, WUS and KAN1 concentrations. A1 (A2) are the static positional cues and $a1c$, $a1k$ and $a2$ their signaling molecules. Parameters $V_x$ represents maximal expression rates (for gene $x$), $k_x$ are the Hill constants, and $n$ the Hill coefficients (always equal to 2). $P_x$ are protein production rates, $g_x$ degradation rates and $D_x$ passive transport rates ($\Delta$ represents the passive diffusion-like transport).

The model and its parameters are described in more detail in Supplementary information. Simulations are performed using a fifth order Runge-Kutta solver with adaptive steps.

In all, 229 model parameter value sets where simulations mirror the biological behavior at equilibrium were obtained with a stochastic gradient descent optimization approach (Supplementary information). The parameter values used for simulations in Figures 5 and 6 are provided in Supplementary Table 8.

In all, 17 perturbations of the system, representing experimental mutants, are studied and modeled by modifying parameters or by adding reactions to the system (Supplementary information). All perturbations are simulated on all the 229 wild-type equilibrium scenarios.

The robustness of the model is assessed using a local parameter sensitivity analysis and in a simulation including cell growth and division (Supplementary information).

Finally, a host of sub-models missing individual regulatory interactions are optimized showing that the model is the minimal model able to reproduce the wild type and perturbations observed in the biological system. Details of the differential equations model, optimization algorithm and model perturbations are given in Supplementary information.

## Data availability

All microarray data used in this study are deposited at http://www.ncbi.nlm.nih.gov/geo/query/acc.cgi?token = fpixfyiwisgumby&acc = GSE29364 (accession number GSE29364).

All simulations were performed using in-house developed software (http://dev.thep.lu.se/organism) and a zip archive containing source code, model files, tissue templates, and all sets of optimized parameter values are available as Supplementary Material or from the web page http://www.thep.lu.se/~henrik/MSB2013/.

## Supplementary information

## Acknowledgements

Live imaging work was done at microscopy core facility of Center for Plant Cell Biology (CEPCEB) and genomics work was done at Institute of Integrative Genome Biology (IIGB), UCR. We thank Yuval Eshed for *6XoP::KAN1*, Thomas Laux for *pWUS::LhG4* transgenic lines and Patricia Springer for *35S::KAN1-GR* transgenic lines and also for comments on manuscript. This work was supported by NSF grant (IOS-1147250) to GVR, and by the Swedish Research Council and the Gatsby Charitable Foundation (HJ). MH acknowledges the Australian Research Council for present funding.

*Author contributions:* RKY, MP and GVR conceived research and designed experiments. RKY and MP performed experiments. JG and HJ developed computational models and performed simulations. RKY, MP, JG, TG, HJ and GVR analyzed the data. CO and MH developed *KAN1:GFP* reporter line used in live imaging experiments. TG helped with microarray analysis. RKY, MP, JG, HJ and GVR wrote the paper. All authors edited and approved the final version of the manuscript.

## Conflict of interest

The authors declare that they have no conflict of interest.

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
