## [Review Process File · Molecular Systems Biology]

Plant stem cell maintenance involves direct transcriptional repression of differentiation program

Ram Kishor Yadav, Mariano Perales, Jérémy Gruel, Carolyn Ohno, Marcus Heisler, Thomas Girke, Henrik Jönsson and G. Venugopala Reddy

Corresponding author: Henrik Jönsson, University of Cambridge

Review timeline:

Submission date:	23 July 2012
Editorial Decision:	09 September 2012
Revision received:	07 December 2012
Editorial Decision:	10 January 2013
Revision received:	30 January 2013
Accepted:	18 February 2013

Editor: Andrew Hufton / Thomas Lemberger

Transaction Report:

1st Editorial Decision

09 September 2012

Thank you again for submitting your work to Molecular Systems Biology. We have now heard back from the three referees who agreed to evaluate your manuscript. As you will see from the reports below, the referees generally found this work of interest. They all had, however, important concerns, which they felt would need to be convincingly addressed before this work would be appropriate for Molecular Systems Biology. These concerns are sufficient that they must preclude publication of this work in its present form.

Importantly, the reviewers indicated that additional experimental work would be needed in some key areas, in particular to support the idea that leaf differentiation factors can repress WUSCHEL.

In addition, I would like to emphasize the points by Reviewer #3 regarding data release. Before you submit any revised work, we ask that submit all new microarray datasets to a public repository, such as GEO or ArrayExpress, and include a confidential reviewer login in the Methods section of the manuscript. Molecular Systems Biology also generally requires that authors provide all new mathematical models in a machine readable common format as supplementary material, and we encourage the use of SBML and deposition to public repositories (e.g. BioModels) whenever appropriate. Please see our Instructions for Authors for more details on our data and model release policies (<http://www.nature.com/msb/authors/>).

Lastly, an issue was raised during review regarding the author contribution statement, which lists Carolyn Ohno and Marcus Heisler only as contributing the KAN1:GFP reporter line. One of the scientists who reviewed this manuscript noted that this would not usually constitute grounds for authorship, and suggested that perhaps the contribution of these authors should be more fully

described. Molecular Systems Biology generally encourages adherence to the ICMJE authorship guidelines (http://www.icmje.org/ethical_1author.html).

If you feel you can satisfactorily deal with these points and those listed by the referees, you may wish to submit a revised version of your manuscript. Please attach a covering letter giving details of the way in which you have handled each of the points raised by the referees. A revised manuscript will be once again subject to review and you probably understand that we can give you no guarantee at this stage that the eventual outcome will be favorable.

Referee reports:

Reviewer #1 (Remarks to the Author):

WUS is expressed in the organizing center of the SAM where it is required for stem cell specification. In this manuscript, Yadav et al use gene expression analysis and a computational model to demonstrate that WUS represses a large number of genes that are expressed in differentiating cells of the SAM. The authors use a WUS inducible system and microarray to identify potential WUS gene targets localized to specific cellular regions of the SAM. ChIP-qPCR was used to show enrichment for WUS at different genomic regions near the KAN1, KAN2, AS2 and YAB3 gene promoters. Fine mapping of WUS binding to these promoters was performed using EMSA and recombinant WUS protein. WUS binding to the KAN1 promoter was supported by supershift assays and repression of a luciferase reporter gene driven by the KAN1 promoter. As a method to study the collective repression of differentiation promoting TFs, a computational model based upon 3-D SAM tissue and the known CLV3-WUS stem cell regulatory network was generated. This model demonstrated a minimal set of interactions for the network to organize the SAM with WUS as the hub with feedback from the CZ and PZ. Transient manipulation of WUS or CLV3 using Dex inducible systems supported a model of WUS-mediated repression of differentiation TFs to block stem cell progenitor differentiation. The authors' concluding gene expression model is a WUS-mediated regulatory network that links the CZ and PZ and maintains the non-differentiated stem cells in the SAM tissue. This study used multiple techniques to elucidate and validate a gene expression model in the SAM which is valuable for advancement in our understanding of cellular differentiation. The manuscript is well organized and well written.

The following are my comments on the manuscript:

1. Figure 1: Missing the scale bar. Panel d, what is the y-axis units and is everything plotted relative to mock? This should be stated somewhere.
2. Figure 2: Gene labels in E & F and G & H are in reverse order, ie: E&F should start with the same gene and end with the same gene. This will make it easier for the reviewer to directly compare the panels.
3. Figure 3: Panel e: EMSA tests an element at KAN1 +1050, however KAN1 ChIP does not go past +800. ChIP past +1050 needs to be performed especially since the element used in the luciferase reporter assay (panel h) is also past +800. This panel also needs + signs in front of the positive values to be consistent with the rest of the figure. The authors should demonstrate an EMSA where at least KAN1, but preferably all elements, do not bind to an oligo with the TAAT sites changed to TGGT. This will strengthen the argument that WUS binds to these sites.
Figure 3: Panel g: please cite the source of the anti-WUS in the methods.
Figure 3: Panel f: The authors need to state the reason for selecting the specific TAAT sequence in each oligo because many of the oligos contain more than one TAAT sequence. The mutant oligos mentioned for panel e would help justify the authors' selection. Please cite the reference for TAAT being a known WUS binding sequence.

4. Figure 4: What are the red, green and blue lines on the panels c-d? Is this a file conversion issue? Missing the scale bar.

5. Figure 5: Missing the scale bar and indication of what arrows represent.

6. Table 1: The legend needs to be shortened

7. Page 6-7: Where are all the other WUS DEGs expressed if they don't map to the CZ, RM or PZ? Can the authors discuss the potential role of these genes? Do they fit into the stem cell maintenance model or are they required for SAM formation or function? When discussing microarray data compared to previous cell sorting, WUS regulated genes should be given as number of genes and not percentages.

8. Page 7-8: WUS represses a cassette of differentiation promoting transcription factors: This entire section should be included in the previous section when discussing the numbers of genes activated and repressed and that section could be changed to this title. I don't feel that this deserves its own section because it is a discussion of the results that were just presented.

9. Page 8: The last sentence states that WUS excludes differentiation promoting TFs from stem cells of the CZ. This statement is too strong and is not supported by the data. The authors lack an experiment demonstrating the exclusion; especially considering that the WUS expression is expanded in the Dex induced system, I feel it would be hard to conclude exclusion.

Minor point: The authors repeatedly use the term "cassette" of genes. This term usually refers to genes that are expressed from the same construct or are adjacent on a chromosome. In this study neither of those is true. Alternatives the authors could use: collection, group, family. The term mis-expression is incorrectly spelled miss-expression in the manuscript and supplemental material.

Reviewer #2 (Remarks to the Author):

Manuscript by Yavad et al, presents solid experimental-theoretical study to reveal minimal regulatory mechanisms underlying stem cell maintenance in plant apical meristems. Authors propose a novel molecular link between WUSCHEL master regulator of stem cell maintenance and downstream transcription factors that control meristem differentiation. They confirm this link experimentally using microarray-based expression and Chip-qPCR data. In line, they build a computer model of the WUSCHEL mediated regulatory network to predict some eminent mutant behaviors. Some of these predictions from the models were further validated by experiments. Although, I found this work interesting and novel I had few concerns that I believe should be clarified before I can recommend this paper for publication in MSB journal:

1. Although authors used inducible WUSCHEL over expression line to reveal repression of leaf differentiation genes (i.e. KAN1). They did not complement these results with WUSCHEL knock-down plants. I believe this is important to yield the more complete understanding of WUSCHEL function. Similar the assumption of KAN1 inhibiting WUSCHEL transcription or activity that was made in the computer model does not seem to be experimentally validated (page 10). Is there any empirical evidence for this feedback loop between WUSCHEL and KAN1-like genes? Authors could elaborate on this issue and discuss the relevance of this particular model assumption for the predicted patterns.

2. Hypothetical activator signals originating from L1 and OC zone seems crucial for the model performance (Supplementary Figure 8) as also shown in their previous studies (i.e. Yavad et al., 2011). Taking into account a large datasets generated to describe shoot meristem maintenance mechanisms, one would ask are those signals biologically plausible? Would they represent for example plant phytohormones? I felt this issue has not been discussed with enough attention in the manuscript.

3. Authors identify 229 parameter sets that explain the 'WT' WUSCHEL-CLV3-KAN1 expression patterns. While going through the supplements, one can find a large-scale analysis of model parameter perturbation has been made to prove qualitative model robustness. However, I could not find which parameters of those 229 are the most important ones. To make life easier for average reader, perhaps authors could summarize the most sensitive parts of the model (parameters) and their biological meaning in form of the table or alternatively clarify this issue in the manuscript. Are those parameters expressed in biologically relevant units (i.e. diffusion of CLV3)? The complexity of their system is enormous therefore, to grab the essence of the proposed mechanism; one would seek for a simplified description.

4. The evidence for KAN1 inhibiting WUSCHEL introduced in the model is vague and based on the observation of meristem collapse phenotype (page 14) but this might not necessary reflect that KAN1 directly inhibits WUSCHEL activity, also model predictions are constrained to the hypothetical assumption. More experiments would be required to identify molecular link for this type of regulation in the network before making a strong conclusion.

5. Throughout the story line, authors claim that over expression of WUSCHEL transcription factor leads to upregulation of CLV3 and severe downregulation of leaf differentiation genes. However, following author's logic CLV3 simultaneously represses WUSCHEL. Therefore, more WUSCHEL could be compensated/balanced by CLV3-mediated repression mechanisms and this could lead to homeostasis in the regulation of leaf differentiation. Since this is a crucial element of the whole story I believe it deserves more elaborated discussion. Would this putative compensation mechanism possibly explain why transient WUSCHEL depletion fails to induce differentiation? (Page 15).

6. Page 13: Sentence starting with: "The clv3 loss-of-function...." Is not clear. How come that in clv3 mutant the CLV3 expression of is increased? I believe this needs some clarification. Following the same page: what mean not a "null-mutant" implementation? This is rather confusing and should be explained in the text.

Reviewer #3 (Remarks to the Author):

This manuscript is an extension of previous work from the same authors and others on the role of WUSCHEL (WUS) as a homeodomain transcription factor. Here the authors provide an molecular explanation on how WUS regulates transcriptionally the stem cell maintenance in the shoot apical meristem. The authors overexpress WUS in an inducible expression system to identify the direct and indirect transcriptional targets. They confirm some of these genes by qPCR and finally they detect the presence of WUS on the promoters of some of the targets. Part of the experimental evidence is used to produce a model that is an updated version from previous modelling efforts from the same authors, adding the some of the new evidence found before. The model can be tuned to robustly show the expression of key genes is similar to the experimental evidence. Perturbations of the network which have experimental counterparts (loss of function, diffusion...) are accurately represented by the model.

I found the manuscript interesting and well written. The experiments described are of high quality. However, there are several points that should be addressed in order to provide a more comprehensive role of WUS in the stem cell niche of the shoot apical meristem, and the impact of the modelling work should be critically assessed:

1. The authors use a Dex inducible system to overexpress WUS and find its potential transcriptional targets. Additionally they look for the direct targets using cycloheximide in the same inducible system. However in the text or tables there is no mention of the overlap between these two datasets and indeed during the results section they were treated as independent sets. Without the lists it is difficult to assess how many direct targets are persistently induced, and how different is the direct response compared to the global downstream transcriptional response.

2. This reviewer had some difficulties to inspect the results obtained by the authors. There is not a link to the raw data (cel files) or identifier from a microarray repository for preprocessing analysis and the Excel files with the tables were corrupted and unavailable for the reviewer, leaving only the

pdfs (21, 31 and 50 MB each) to check the values. That poses a problem in order to provide any constructive criticism that should be corrected. Also it would help to have a list in the supplementary tables with only the list of DEGs, for example, with the 641 DEGs positively regulated after Dex.

3. Within the DEGs in Dex only, ~40% are upregulated and ~60% are downregulated, and a similar trend is observed for Cyc (33% upregulated, 66% downregulated). These numbers indicate that WUS has a role as a transcriptional repressor but also as an activator. However, the next sections in the manuscript focus exclusively on the role of WUS as a repressor of a small subset of genes ignoring the global effect on the rest of the genes affected. Even though it is a very interesting finding (the repression of key components of leaf differentiation as KAN1, KAN2, AS2 and YAB3), there is no further follow-up on the role of WUS on the upregulated genes or the rest of the downregulated genes. For example, there is no analysis of any of the sets of targets that could reveal transcriptional modules regulated by WUS (others than leaf polarity establishment and differentiation).

4. By their focus on leaf developmental genes, the authors stress a role for WUS in the repression of differentiation as an analogy to the way in which key stem cell factors in animal pluripotent stem cells keep 'differentiation genes' silent. I feel that this comparison is not justified, as it seems that many of the repressed genes also play roles in patterning processes in the shoot, without being clearly analogous to animal cell lineage commitment factors.

5. In a similar way, the authors indicate that 38% of the downregulated DEGs in Dex only are expressed in the CZ/RM zone, while 10% are expressed in PZ/RM (for Cyc 29% PZ/RM and 11% CZ/RM). These numbers are far from a major trend in the sets, indeed most of the genes are expressed in all or in none of the three cell types. In the case of the upregulated these trends are even smaller (17% CZ/RM, 5% PZ Dex only, 17% CZ/RM, 8% PZ/RM). However, the authors focus on some of the best understood genes to characterize the transcriptional response.

6. The ChIP experiment revealed the presence of a box that is common to four targets. A simple bioinformatics analysis (e.g. MEME) would confirm the presence of such a motif in the promoter of most of the potential direct targets.

7. It is not easy for me to understand from the manuscript whether the computational model is really giving us new insights. The framework of the model has been published in previous papers that focus on the positioning of the CLV3 and WUS domains. Now, the authors have added the differentiation factors, incorporating that they are repressed by WUS and assuming (but not proving directly) that they in turn repress WUS, as well as assuming some other unproven inputs. In my opinion it is trivial that the reported network leads to central WUS and peripheral KAN1 expression, and as such it does not appear to significantly contribute to the paper.

8. Small comments on the model: it is not clear to me which assumptions are made so KAN1 is not expressed in the CLV3 domain.

9. The authors state: " Thus far, our analysis suggests that stem cell maintenance may be a systems property that arises as a result of collective repression of differentiation promoting transcription factors...". However, the model is quite deterministic about the initial position of all the components and the signals that trigger them, so it is difficult to see it as a "systems property" rather than as an effect of their deterministic model.

Reviewer #1 (Remarks to the Author):

WUS is expressed in the organizing center of the SAM where it is required for stem cell specification. In this manuscript, Yadav et al use gene expression analysis and a computational model to demonstrate that WUS represses a large number of genes that are expressed in differentiating cells of the SAM. The authors use a WUS inducible system and microarray to identify potential WUS gene targets localized to specific cellular regions of the SAM. ChIP-qPCR was used to show enrichment for WUS at different genomic regions near the KAN1, KAN2, AS2 and YAB3 gene promoters. Fine mapping of WUS binding to these promoters was performed using EMSA and recombinant WUS protein. WUS binding to the KAN1 promoter was supported by supershift assays and repression of a luciferase reporter gene driven by the KAN1 promoter. As a method to study the collective repression of differentiation promoting TFs, a computational model based upon 3-D SAM tissue and the known CLV3-WUS stem cell regulatory network was generated. This model demonstrated a minimal set of interactions for the network to organize the SAM with WUS as the hub with feedback from the CZ and PZ. Transient manipulation of WUS or CLV3 using Dex inducible systems supported a model of WUS-mediated repression of differentiation TFs to block stem cell progenitor differentiation. The authors' concluding gene expression model is a WUS-mediated regulatory network that links the CZ and PZ and maintains the non-differentiated stem cells in the SAM tissue. This study used multiple techniques to elucidate and validate a gene expression model in the SAM, which is valuable for advancement in our understanding of cellular differentiation. The manuscript is well organized and well written.

The following are my comments on the manuscript:

1. Figure 1: Missing the scale bar. Panel d, what is the y-axis units and is everything plotted relative to mock? This should be stated somewhere.

Author response: We thank the reviewer for this comment. We have now provided the scale bar in panel C which remains same for both B and C, and this aspect is now mentioned in the legend. Error bars in D represent standard deviation derived from two biological replicates, which is now mentioned in the legend. y-axis label has been updated which represents the fold change in Dex-treated samples relative to Mock-treated samples.

2. Figure 2: Gene labels in E & F and G & H are in reverse order, ie: E&F should start with the same gene and end with the same gene. This will make it easier for the reviewer to directly compare the panels.

Author response: We thank the reviewer for noticing this, and Fig. 2 has been updated to address reviewer's concern.

3. Figure 3: Panel e: EMSA tests an element at KAN1 +1050, however KAN1 ChIP does not go past +800. ChIP past +1050 needs to be performed especially since the element used in the luciferase reporter assay (panel h) is also past +800. This panel also needs + signs in front of the positive values to be consistent with the rest of the figure. The authors should demonstrate an EMSA where at least KAN1, but preferably all elements, do not bind to an oligo with the TAAT sites changed to TGGT. This will strengthen the

argument that WUS binds to these sites.

Author response: We fully agree with the reviewer, and as suggested, ChIP-qPCR analysis has now been extended further for KAN1 and KAN2 genes. Moreover all positive (+) signs were removed from this figure to be consistent with EMSA analyses. Additionally we present in a new Figure 4, a complete mutagenesis analysis of WUS binding elements reinforcing the WUS binding preferences for TAAT elements is specific. A description of these results was added to the main text on page 9 (last line) and 10, and in an addition in the Discussion page 20 first paragraph.

Figure 3: Panel g: please cite the source of the anti-WUS in the methods.

Author response: We thank the reviewer for this comment. Source of the antibody (Yadav et al., *Genes Dev.* 25: 2025-2030) has been cited in methods section (page 25).

Figure 3: Panel f: The authors need to state the reason for selecting the specific TAAT sequence in each oligo because many of the oligos contain more than one TAAT sequence. The mutant oligos mentioned for panel e would help justify the authors' selection. Please cite the reference for TAAT being a known WUS binding sequence.

Author response: We thank the reviewer for this comment. Mutational analyses of WUS binding oligonucleotides have been carried out to determine the specific binding of WUS for TAAT containing sequences. An additional figure (Figure 4) has been provided showing the results of these analyses which supports the selective binding of WUS for TAAT containing elements. Earlier work showing WUS binding to TAAT containing sequences has been cited (page 9 and 10).

4. Figure 4: What are the red, green and blue lines on the panels c-d? Is this a file conversion issue? Missing the scale bar.

Author response: We thank the reviewer for spotting these lines coming from the image visualization software. The figure has been corrected to remove these lines. (new Figure 5)

5. Figure 5: Missing the scale bar and indication of what arrows represent.

Author response: We thank the reviewer for this comment. Scale bar provided in panel B remains the same for all panels. Arrows point to cells expressing KAN1:GFP (nuclear expression). This is now described in the figure legend (new Figure 6).

6. Table 1: The legend needs to be shortened

Author response: We thank the reviewer for this suggestion. The legend has been shortened.

7. Page 6-7: Where are all the other WUS DEGs expressed if they don't map to the CZ, RM or PZ? Can the authors discuss the potential role of these genes? Do they fit into the

stem cell maintenance model or are they required for SAM formation or function? When discussing microarray data compared to previous cell sorting, WUS regulated genes should be given as number of genes and not percentages.

Author response: We thank the reviewer for this comment. We have now included actual numbers in the text along with percentages. About two third of WUS activated genes do not map to any one of the three cell types. Similarly, out of 303 WUS repressed genes only 132 genes map to the three cell types. This would imply that WUS activates or represses a subset of genes that may be broadly expressed in SAMs, which is conceivable given the broader domain in which the WUS protein is detected. Alternatively, the resolution of the expression map involving only three cell types may be limited. For example, *CLV3* expression domain overlaps with that of *WUS*, which limits separation of genes enriched in individual cell layers of the SAM (Yadav et al., 2009, *Proc Natl Acad Sci USA* 106: 4941-4946). We have developed new markers to increase the resolution of expression map both along the radial domain and across different cell layers. The mapping of WUS-responsive genes to this high-resolution map, upon completion, will provide a better cell type specific representation of WUS responsive genes. We have now added this discussion to “WUS represses a cassette of differentiation promoting transcription factors” – Pages 6 and 7.

The gene ontology (GO) analysis of WUS responsive genes has been provided in the revised manuscript (pages 17-18). Important developmental regulators that are activated by WUS include *CUP SHAPED COTYLEDON1 (CUC1)* that is involved in SAM formation and maintenance, *EXCESS MICROSPOROCTES1 (EMS1)*, a receptor like kinase that promotes cytokinesis during anther development and ARR4-type-A response regulator implicated in negatively regulating cytokinin signaling and *GLABRA1* implicated in epidermal cell fate specification. Although candidates such as *CUC1* are relevant in SAM function, a high resolution analysis of their function is required to include them in our stem cell regulatory model. WUS down regulates genes involved in indole derivatives biosynthesis process (GO:0042435, $p < 8.19 \times 10^{-5}$). This is interesting since auxin has been shown to promote differentiation of leaves in SAMs and differentiation of daughters of columella initials in the root. Again functional analysis of WUS repressed genes such as *ANTHRANILATE SYNTHASE ALPHA SUBUNIT1 (ASA1)*, *ASA1* acts as a rate-limiting step in tryptophan biosynthesis, *TRYPTOPHAN AMINOTRANSFERASE RELATED2 (TAR2)*, encodes a protein similar to *TRYPTOPHAN AMINOTRANSFERASE OF ARABIDOPSIS1* involved in local IAA biosynthesis and *PHYTOALEXIN DEFICIENT3 (PAD3)*, *PAD3* which encodes a cytochrome P450 enzyme that catalyzes the conversion of dihydrocamalexin acid to indole derivative camalexin is required to incorporate them into the existing stem cell regulatory model. Therefore, we discuss in the manuscript that WUS-mediated activation of *CLV3* and repression of differentiation promoting transcription factors is a minimal network that can explain stem cell homeostasis. This analysis and discussion has been described under new section “Other WUS responsive gene functions and pathways” on pages 17 and 18.

8. Page7-8: WUS represses a cassette of differentiation promoting transcription factors: This entire section should be included in the previous section when discussing the numbers of genes activated and repressed and that section could be changed to this title. I don't feel that this deserves its own section because it is a discussion of the results that were just presented.

Author response: We thank the reviewer for this suggestion. We have rearranged this section as suggested (pages 6-9).

9. Page 8: The last sentence states that WUS excludes differentiation-promoting TFs from stem cells of the CZ. This statement is too strong and is not supported by the data. The authors lack an experiment demonstrating the exclusion; especially considering that the WUS expression is expanded in the Dex induced system, I feel it would be hard to conclude exclusion.

Author response: We agree with the reviewer and the sentence has been modified (page 10, last sentence).

Minor point: The authors repeatedly use the term "cassette" of genes. This term usually refers to genes that are expressed from the same construct or are adjacent on a chromosome. In this study neither of those is true. Alternatives the authors could use: collection, group, family.

Author response: We agree with the reviewer and term cassette has been removed throughout the manuscript.

The term mis-expression is incorrectly spelled miss-expression in the manuscript and supplemental material.

Author response: We thank the reviewer for spotting this and we have corrected this mistake throughout the manuscript and supplementary information.

Reviewer #2 (Remarks to the Author):

Manuscript by Yadav et al, presents solid experimental-theoretical study to reveal minimal regulatory mechanisms underlying stem cell maintenance in plant apical meristems. Authors propose a novel molecular link between WUSCHEL master regulator of stem cell maintenance and downstream transcription factors that control meristem differentiation. They confirm this link experimentally using microarray-based expression and Chip-qPCR data. In line, they build a computer model of the WUSCHEL mediated regulatory network to predict some eminent mutant behaviors. Some of these predictions from the models were further validated by experiments. Although, I found this work interesting and novel I had few concerns that I believe should be clarified before I can recommend this paper for publication in MSB journal:

1. Although authors used inducible WUSCHEL over expression line to reveal repression

of leaf differentiation genes (i.e. KAN1). They did not complement these results with WUSCHEL knock-down plants. I believe this is important to yield the more complete understanding of WUSCHEL function. Similar the assumption of KAN1 inhibiting WUSCHEL transcription or activity that was made in the computer model does not seem to be experimentally validated (page 10). Is there any empirical evidence for this feedback loop between WUSCHEL and KAN1-like genes? Authors could elaborate on this issue and discuss the relevance of this particular model assumption for the predicted patterns.

Author response: We thank the reviewer for this comment. We have tried to inducibly knock down WUS by using dex inducible systems driving RNAi constructs as shown our earlier manuscript (Yadav et al., *Development* 137:3581-3589) but we find it very difficult to maintain these WUS RNAi phenotypes in dex-inducible manner over generations. We tried these experiments again in the last couple of months but we were not successful in generating stable Dex-inducible WUS RNAi lines. Therefore as an alternate system, we have used inducible overexpression of CLV3 to down regulate WUS levels (Fig. 6C and D) and now provide new data to show that inducible activation of CLV3 results in downregulation of *WUS* transcript levels (Supplementary Figure 8). Similar CLV3 overexpression systems have been used in earlier studies to down regulate WUS levels to characterize WUS response genes [Leibfried et al., *Nature*. 2005 Dec 22;438(7071):1172-5].

With respect to KAN1 inhibiting WUS expression/activity. This assumption was obtained from an earlier study where it has been documented that ubiquitous overexpression of KAN1 leads to SAM growth arrest (Kerstetter R.A. et al., 2001, *Nature* 411: 706-708). We have now used the Dex-inducible system to ubiquitously overexpress KAN1 to show that KAN1 overexpression leads to downregulation of WUS transcript levels (Supplementary Figure 5). This aspect of the model assumption is further discussed under point #4.

2. Hypothetical activator signals originating from L1 and OC zone seems crucial for the model performance (Supplementary Figure 8) as also shown in their previous studies (i.e. Yadav et al., 2011). Taking into account a large datasets generated to describe shoot meristem maintenance mechanisms, one would ask are those signals biologically plausible? Would they represent for example plant phytohormones? I felt this issue has not been discussed with enough attention in the manuscript.

Author response: We thank the reviewer for this comment. We agree with the reviewer that spatial signals in superficial cell layers of the CZ may play an important role in confining CLV3 expression to only the CZ cells despite detection of WUS protein in much broader domain that encompasses the RM, the CZ and parts of the PZ. One such spatial signal could be the local synthesis of active cytokinins in superficial cell layers as suggested by the expression patterns of LONELYGUY class of genes [Yadav et al., *Proc Natl Acad Sci USA* 106: 4941-4946. 2009; Chickarmane VS, *Proc Natl Acad Sci U S A*. 2012 Mar 6;109(10):4002-7]. This aspect has been added to the discussion on page 20.

Since there is no direct molecular evidence on an L1 originating signal for activating

CLV3, we have tried different possibilities in computational models for generating the domain asymmetrically outside the *WUS* domain (*eg.* repression from the stem or PZ or activation from the top of the meristem), which is partly covered in previous modeling efforts [Jonsson et al 2003, Yadav et al 2011]. In conclusion, the choice of this activator is highly motivated by the *pCLV3::WUS* mutant, showing an expression of both *CLV3* and the transfected *WUS* in the first three layers of the meristem. All the options we have tested but the *WUS/L1* activation fail to reproduce this case without additional hypotheses.

The signal in the model from L1 to activate *KANI* is only motivated by the expression pattern seen in the confocal data. In models not including this interaction, the PZ gene is expressed also in more internal layers (in the PZ), but no conclusions from the modeling work would be altered.

The *WUS* activation from an OZ located signal is motivated by cytokinin binding to *AHK4* receptors (located in and around the OC). This binding activates the downstream signalling cascade leading to the transcriptional activation of *WUS* [Chickarmane VS, Proc Natl Acad Sci U S A. 2012 Mar 6;109(10):4002-7]. An earlier modeling study has also shown that a localized *WUS* activation is important to describe some dynamic perturbations [Jonsson et al, 2005].

We have extended the discussion of these aspects of the model in the main text when defining the model (page 11-12) and in the Supplementary Information and we have added a comment on the possibility of the L1 signal in the discussion (page 20).

3. Authors identify 229 parameter sets that explain the 'WT' *WUSCHEL-CLV3-KANI* expression patterns. While going through the supplements, one can find a large-scale analysis of model parameter perturbation has been made to prove qualitative model robustness. However, I could not find which parameters of those 229 are the most important ones. To make life easier for average reader, perhaps authors could summarize the most sensitive parts of the model (parameters) and their biological meaning in form of the table or alternatively clarify this issue in the manuscript. Are those parameters expressed in biologically relevant units (*i.e.* diffusion of *CLV3*)?

The complexity of their system is enormous therefore, to grab the essence of the proposed mechanism; one would seek for a simplified description.

Each one of the 229 parameter sets is able to reproduce the *wild type* gene expression patterns. All of them also produce a behavior that, qualitatively, is able to reproduce experimentally observed perturbations. From that point of view, one cannot consider one parameter set as more biologically relevant than another. We use this distribution of parameter values to show that the model behavior is robust across large variations of parameter values. Also, the quantitative variation in model behavior across parameter sets could possibly reflect response variations in a population of plants. The discussion, page 20 (last paragraph) of the main text, elaborates on this point.

The model, mostly based on qualitative molecular data, is not expressed in biologically relevant units. For instance, the model features a repression of *WUS* by

clv3, this repression, modeled with two parameters of a Hill function, represents the binding of clv3 to its receptor, the activation of the downstream signaling pathway, the binding of transcription factors to *WUS* promoter, and including all these steps would result in an increase of number of model parameters without higher resolution in the comparison with data. At the present time, data is too scarce and the system too large to accurately represent the complete molecular dynamics of the system, and use biologically relevant units, but still relative values among parameters can be indicated (e.g. the combination of *WUS* production/degradation/intercellular transport rate must be set such that the *WUS* gradient becomes correct). In conclusion, different parts of the model are important for solving different tasks, as has been discussed in more detail in our earlier modeling work, but to solve all current comparisons with data, all interactions are necessary, i.e. it is a network/systems property.

In order to facilitate the understanding of the differential equation system, we have briefly extended the parameter/mechanism discussion in the main text (page 16) and the model section of the Supplementary Information text, describing the system interactions and parameters, has been modified and features a more elaborate description of parameters biological meaning.

4. The evidence for KAN1 inhibiting WUSCHEL introduced in the model is vague and based on the observation of meristem collapse phenotype (page 14) but this might not necessary reflect that KAN1 directly inhibits WUSCHEL activity, also model predictions are constrained to the hypothetical assumption. More experiments would be required to identify molecular link for this type of regulation in the network before making a strong conclusion.

Author response: We thank the reviewer for this comment. As stated under point#1, we have provided new data which shows downregulation of *WUS* transcript levels upon ubiquitous overexpression of KAN1 (Supplementary Figure 5). We agree with the reviewer that KAN1 inhibition of *WUS* expression probably is indirect due to the non complementarity of their expression patterns. This indirect negative effect could be due to the abaxialization of developing leaves upon KAN1 misexpression and abaxialization of leaves correlates with SAM growth arrest. The indirect and negative effects of differentiating cells on the central pattern (*WUS* and *CLV3*) has been documented in a recent study [Goldshmidt et al., *Plant Cell*. 2008 May;20(5):1217-30]. This study reveals that down regulation of two YABBY class of genes-*YABBY3* and *FILAMENTOUS FLOWER (FIL)* leads to expansion of *WUS* promoter activity suggesting that yet to be identified signal derived from differentiating cells inhibits *WUS* expression, organizes the central patterns and SAM growth. Interestingly we find *YAB3* is also one of the genes directly repressed by *WUS*. Therefore, the model assumption of KAN1 inhibiting *WUS* expression should be viewed as PZ-derived signal inhibiting *WUS* expression and this aspect has been restated in the revised manuscript (page 12). Since the identity of the PZ-derived signal is elusive, it is modeled as having direct negative effect on *WUS* transcription. From the modeling perspective, the negative regulation of *WUS* by KAN1 is not essential to reproduce wild type expression domains and also the

domain reorganizations observed upon perturbations in CLV3 or WUS levels (this can for example be seen in the perturbation analysis (Supplemental Figure 10), where altering KAN parameters have no effect on CLV3 and WUS). Given the data for overexpression of KAN1, we still think it is interesting to add the interaction from KAN1 to the core CLV3/WUS network, and the only way we can reproduce the KAN1 perturbation experiments is if it is acting repressively (although indirect) on WUS (or on the WUS activation). To avoid confusion we have indicated this directly in Figure 5A and added a comment on this in the main text (page 12). In addition we have added a more detailed analysis for different scenarios for KAN1 feeding into the CLV3/WUS network in the Supplementary Information (“KAN1 effect on the WUS-CLV3 network”).

5. Throughout the story line, authors claim that over expression of WUSCHEL transcription factor leads to upregulation of CLV3 and severe downregulation of leaf differentiation genes. However, following author's logic CLV3 simultaneously represses WUSCHEL. Therefore, more WUSCHEL could be compensated/balanced by CLV3-mediated repression mechanisms and this could lead to homeostasis in the regulation of leaf differentiation. Since this is a crucial element of the whole story I believe it deserves more elaborated discussion. Would this putative compensation mechanism possibly explain why transient WUSCHEL depletion fails to induce differentiation? (Page 15).

Author response: We thank the reviewer for this important comment which has allowed us to discuss further the WUS-mediated regulation of linking the CZ and the PZ through a direct transcriptional regulation to achieve homeostasis balance between the different SAM domains. As shown in the new data provided, both in vivo WUS and CLV3 transcript levels are only downregulated but not completely terminated upon dex-inducible overactivation of *CLV3* (Supplementary Figure 8) suggesting a dynamic balance between the levels of these two genes. In the model we have used strong perturbations for all overactivation mutants following our earlier studies, but now also provide a figure (Supplementary Figure 10) and comment upon this in the main text (page 14). Transient perturbations will in general return to the wild-type expression patterns when the perturbation is removed, since over a large range of initial conditions the model will generate these. Also, the model captures the dynamic homeostasis between CLV3 and WUS by lowering the effect of their individual perturbations, now discussed in the main text (page 16). Thus the WUS-mediated repression of differentiation program observed in this study links the PZ with well-established feedback loop between cells of the CZ and the RM through a direct transcriptional control. The dual function of WUS in activating its own negative regulator and repressing differentiation would lead to a mechanism in balancing stem cell numbers and differentiation rates of stem cell progenitors. This apart, an earlier study have also shown that WUS levels also influence cell division rates of stem cell daughters within the PZ (Yadav et al., *Development* 137:3581-3589). Though at this point we do not know how WUS regulates cell division, however, a single factor controlling its own levels (through CLV3 activation in the CZ), repression of differentiation in the PZ and cell division rates in the PZ should lead to a robust mechanism in maintaining the balance between stem cell numbers and differentiation rates. These aspects have now been

discussed in the first paragraph of the discussion (page 18 and 19). We agree with the reviewer that this dynamic restoration of WUS levels might have prevented appearance of differentiation markers in the CZ and this point has been included in the discussion.

6. Page 13: Sentence starting with: "The *clv3* loss-of-function.... "Is not clear. How come that in *clv3* mutant the CLV3 expression of is increased? I believe this needs some clarification. Following the same page: what mean not a "null-mutant " implementation? This is rather confusing and should be explained in the text.

We thank the reviewer for the comments and our apologies for the lack of clarity. The first sentence should be stated as CLV3 promoter activity and not CLV3 expression. (loss-of-function has been implemented at the protein/peptide level such that we can follow the promoter activity of the perturbed gene in the model). This sentence has been modified in the revised manuscript (page 15).

By “not a null-mutant implementation”, we imply that the mutation was not absolute. In the case of *WUS loss-of-function*, some WUS proteins are still present in the system – but at a much lower level than in the wild type. In a model of a null mutant mutation, we would have completely removed WUS, resulting in a stronger phenotype. The main text has been amended to clarify this (page 16).

Reviewer #3 (Remarks to the Author):

This manuscript is an extension of previous work from the same authors and others on the role of WUSCHEL (WUS) as a homeodomain transcription factor. Here the authors provide a molecular explanation on how WUS regulates transcriptionally the stem cell maintenance in the shoot apical meristem. The authors overexpress WUS in an inducible expression system to identify the direct and indirect transcriptional targets. They confirm some of these genes by qPCR and finally they detect the presence of WUS on the promoters of some of the targets. Part of the experimental evidence is used to produce a model that is an updated version from previous modelling efforts from the same authors, adding the some of the new evidence found before. The model can be tuned to robustly show the expression of key genes is similar to the experimental evidence. Perturbations of the network which have experimental counterparts (loss of function, diffusion...) are accurately represented by the model.

I found the manuscript interesting and well written. The experiments described are of high quality. However, there are several points that should be addressed in order to provide a more comprehensive role of WUS in the stem cell niche of the shoot apical meristem, and the impact of the modelling work should be critically assessed:

1. The authors use a Dex inducible system to overexpress WUS and find its potential

transcriptional targets. Additionally they look for the direct targets using cycloheximide in the same inducible system. However in the text or tables there is no mention of the overlap between these two datasets and indeed during the results section they were treated as independent sets. Without the lists it is difficult to assess how many direct targets are persistently induced, and how different is the direct response compared to the global downstream transcriptional response.

Author response: We thank the reviewer for this suggestion and we have now provided the overlap between WUS response genes (dex treatment) and potential direct transcriptional target genes (dex+cycloheximide treatment) by constructing a four-way Venn diagram (Supplementary Figure 1). A comparison of Dex-treated samples with mock identified 641 genes as differentially expressed [DEGs] ($\geq/\leq 2$ fold; $p < 0.01$) which consisted of 238 upregulated genes and 403 downregulated genes. While a comparison of Dex+Cyc with Cyc identified 457 genes as DEGs ($\geq/\leq 2$ fold; $p < 0.01$) which consisted of 154 upregulated genes and 303 downregulated genes (Supplementary Table 2). Next, we compared the overlap between genes that respond to WUS upon Dex treatment alone with and Dex and Cyc treatment revealed a common set of only 49 upregulated genes and 140 downregulated genes (Supplementary Figure 1). This additional analysis has been added to page 6 in the manuscript.

2. This reviewer had some difficulties to inspect the results obtained by the authors. There is not a link to the raw data (cel files) or identifier from a microarray repository for preprocessing analysis and the Excel files with the tables were corrupted and unavailable for the reviewer, leaving only the pdfs (21, 31 and 50 MB each) to check the values. That poses a problem in order to provide any constructive criticism that should be corrected. Also it would help to have a list in the supplementary tables with only the list of DEGs, for example, with the 641 DEGs positively regulated after Dex.

Author response: Our apologies for not providing link to raw microarray data. We have now provided a link for all 12 datasets (<http://www.ncbi.nlm.nih.gov/geo/query/acc.cgi?token=fpixfyiwisgumby&acc=GSE29364>).

We again apologize for improper conversion of Excel files during the submission process. We have now provided the Excel files and Supplementary Table 1 should provide all necessary analysis to determine DEGs across all treatments.

In addition as requested by the reviewer, we have now provided two additional tables - Supplementary Tables 2 and 3 that provide lists of up and down regulated genes in dex and dex plus cyc treatments respectively.

3. Within the DEGs in Dex only, ~40% are upregulated and ~60% are downregulated, and a similar trend is observed for Cyc (33% upregulated, 66% downregulated). These

numbers indicate that WUS has a role as a transcriptional repressor but also as an activator. However, the next sections in the manuscript focus exclusively on the role of WUS as a repressor of a small subset of genes ignoring the global effect on the rest of the genes affected. Even though it is a very interesting finding (the repression of key components of leaf differentiation as KAN1, KAN2, AS2 and YAB3), there is no further follow-up on the role of WUS on the up-regulated genes or the rest of the down-regulated genes. For example, there is no analysis of any of the sets of targets that could reveal transcriptional modules regulated by WUS (others than leaf polarity establishment and differentiation).

Author response: We thank reviewer for this comment. The observation made by reviewer that WUS act not only as a repressor but also as an activator in both Dex and Dex+Cyc is noteworthy. We have now carried out Gene Ontology analysis to identify to infer biological functions of DEGs other than those that have been characterized experimentally in the manuscript.

The gene ontology (GO) analysis of WUS responsive genes has been provided in the revised manuscript (Supplementary Table 6). The statistically significant GO categories of WUS up-regulated transcriptome included developmental process (GO:0032502; $p < 0.05$), cell fate commitment (GO:0001708; $p < 0.01$) and response to abscisic acid (GO:0009737; $p < 0.01$). Important developmental regulators that are activated by WUS include *CUP SHAPED COTYLEDON1 (CUC1)* that is involved in SAM formation and maintenance, *EXCESS MICROSPOROCYTES1 (EMS1)*, a receptor like kinase that promotes cytokinesis during anther development and ARR4-type-A response regulator implicated in negatively regulating cytokinin signaling and *GLABRA1* implicated in epidermal cell fate specification. Though candidates such as *CUC1* are relevant in SAM function, however, a high resolution analysis of their function is required to include them in stem cell regulatory model.

The statistically significant GO categories of WUS up-regulated transcriptome included indole derivatives biosynthesis process (GO:0042435, $p < 8.19 \times 10^{-5}$), regulation of gene expression (GO:0010468, $p < 0.002$), post-translation modification (GO:0043687, $p < 0.01$) and organ polarity specification (GO:0009944, $p < 0.03$). The indole biosynthesis process related genes included *ANTHRANILATE SYNTHASE ALPHA SUBUNIT1 (ASA1)*, which catalyzes a rate-limiting step in tryptophan biosynthesis, *TRYPTOPHAN AMINOTRANSFERASE RELATED2 (TAR2)* which encodes a protein similar to *TRYPTOPHAN AMINOTRANSFERASE OF ARABIDOPSIS1* involved in local IAA biosynthesis and *PHYTOALEXIN DEFICIENT3 (PAD3)*, which encodes a cytochrome P450 enzyme that catalyzes the conversion of dihydrocamalexin acid to indole derivative camalexin. This is interesting since auxin has been shown to promote differentiation of leaves in SAMs and differentiation of daughters of columella initials in the root. Again functional analysis of WUS repressed genes such as is required to incorporate them into the existing stem cell regulatory model. Since repression of differentiation promoting transcription factor stood out, a few of them have been selected for further biochemical and functional analysis in current study. Therefore, we have mentioned

in the manuscript that WUS-mediated activation of *CLV3* and repression of differentiation promoting transcription factors is a minimal network that can explain stem cell homeostasis.

This analysis and discussion has been described under new section “Other WUS responsive gene functions and pathways” on pages 17 and 18.

4. By their focus on leaf developmental genes, the authors stress a role for WUS in the repression of differentiation as an analogy to the way in which key stem cell factors in animal pluripotent stem cells keep 'differentiation genes' silent. I feel that this comparison is not justified, as it seems that many of the repressed genes also play roles in patterning processes in the shoot, without being clearly analogous to animal cell lineage commitment factors.

Author response: We thank the reviewer for this comment. Note that WUS besides repressing TFs involved in early aspects of leaf differentiation, it also represses TFs involved in later aspects of cell fate commitment in leaves such as stomatal and vascular differentiation suggesting that WUS potentially binds to regulatory regions of these genes and represses their expression in SAMs. Therefore we have argued that the general logic of repression of differentiation promoting TFs through a direct transcriptional regulation is conserved while the identity of differentiation promoting TFs may be different in both animal and plant kingdoms. This is also true for several analogous processes where the developmental logic seems to be conserved between animals and plants. For example, homeotic control of flower patterning involves MADS domain TFs as opposed to HOX domain TFs that provide segmental identity in animal systems. Similarly plant stem cell promoting TF-WUS (a homeodomain TF), and animal stem cell promoting TFs- OCT4 (POU domain TF) and SOX2 (High Mobility Group DNA binding domain TF) belong to different TF families but carry out analogous functions. Therefore we have modified the abstract which now states that “logic” is conserved (page2).

5. In a similar way, the authors indicate that 38% of the downregulated DEGs in Dex only are expressed in the CZ/RM zone, while 10% are expressed in PZ/RM (for Cyc 29% PZ/RM and 11% CZ/RM). These numbers are far from a major trend in the sets, indeed most of the genes are expressed in all or in none of the three cell types. In the case of the upregulated these trends are even smaller (17% CZ/RM, 5% PZ Dex only, 17% CZ/RM, 8% PZ/RM). However, the authors focus on some of the best understood genes to characterize the transcriptional response.

Author response: We thank the reviewer for this comment. We have now included actual numbers in the text along with percentages. About two third of WUS activated genes do not map to any one of the three cell types. Similarly, out of 303 WUS repressed genes only 132 genes map to the three cell types. This would imply that WUS activates or represses a subset of genes that may be broadly expressed in SAMs, which is conceivable given the broader domain in which the WUS protein is

detected. Alternatively, the resolution of the expression map involving only three cell types may be limited. For example, *CLV3* expression domain overlaps with that of WUS which limits separation of genes enriched in individual cell layers of the SAM (Yadav et al., 2009, *Proc Natl Acad Sci USA* 106: 4941-4946). We have developed new markers to increase the resolution of expression map both along the radial domain and across different cell layers. The mapping of WUS-responsive genes to this high-resolution map, upon completion, will provide a better cell type specific representation of WUS responsive genes.

We have now added this discussion to “WUS represses a group of differentiation promoting transcription factors” – Pages 6 and 7, and concluded as follows “Taken together these results reveal that a majority of WUS activated genes are broadly expressed within the central parts of SAMs and represses a large group of genes expressed in the PZ” on page 8.

6. The ChIP experiment revealed the presence of a box that is common to four targets. A simple bioinformatics analysis (e.g. MEME) would confirm the presence of such a motif in the promoter of most of the potential direct targets.

Author response: We thank the reviewer for this suggestion. Note that the new mutational analysis of WUS binding sequences presented in Figure 4 shows that WUS binds to sequences containing “TAAT” core, however, not all “TAAT” sequences bind WUS suggesting a requirement of some additional contextual information. Therefore, a simple bioinformatics approach will yield several “TAAT” sequences all of which may not be relevant. The best solution is to obtain a genome wide WUS binding sequences. In fact we tried ChIP-seq analysis but found out that the data is too noisy to deduce WUS binding patterns. An earlier study which attempted to deduce genome wide WUS binding reported that about 93% of genes bound by WUS are not regulated by WUS and 99% of WUS-regulated genes are not bound by WUS [Busch W., et al., 2010, *Dev Cell* 18: 849-861; Clark SE, 2010, *Dev Cell*. 18(5):696-7]. While biological reasons are possible to explain this disparity, a simple explanation could be that the genome wide methods are yet to be standardized to obtain good quality data to deduce WUS binding cis-element code.

Note that a comparison of “TAAT” containing WUS binding sequences that we have characterized so far (Figures 3 and 4) do not reveal any other pattern associated with it. This also suggests an alternate model wherein three dimensional shape of binding sequence may determine WUS binding specificity, similar to what has been shown in the case of *Drosophila* HOX protein-DNA recognition wherein the minor groove width determines binding specificity [Rohs et al., 2009, *Nature*. 461(7268):1248-53; Slattery et al., 2011, *Cell*. 2011, 147(6):1270-82]. Therefore, in the absence of genome wide binding patterns, we have characterized promoters of individual genes to find sequences that bind WUS and provided only a minimal network that is sufficient to explain stem cell maintenance.

Part of this discussion is added to pages 20.

7. It is not easy for me to understand from the manuscript whether the computational model is really giving us new insights. The framework of the model has been published in previous papers that focus on the positioning of the CLV3 and WUS domains. Now, the authors have added the differentiation factors, incorporating that they are repressed by WUS and assuming (but not proving directly) that they in turn repress WUS, as well as assuming some other unproven inputs. In my opinion it is trivial that the reported network leads to central WUS and peripheral KAN1 expression, and as such is does not appear to significantly contribute to the paper.

Author response: We thank the reviewer for this comment. We have now provided the data to show that ubiquitous overexpression of KAN1 has negative impact on WUS expression (Supplementary Figure 5). The negative inhibition of KAN1 on WUS and other interactions has been explained at various places to address comments from reviewer 2. The model presented in this paper is indeed an extension of previously published models, as pointed out in manuscript and supplementary information, but it is the first model, to our knowledge, where WUS is given multiple tasks to activate CLV3 expression in a negative feedback loop and repress genes in PZ. The fact that the suggested model is sufficient in explaining three-dimensional expression patterns as well as dynamical perturbations is in our opinion not trivial to understand by intuition alone.

Also, the work proposed here offers an in depth analysis of the model, showing the robustness of the proposed network given the variety of parameter values showed to behave qualitatively the same. We propose an analysis of 17 perturbations of the system, extensively compared with the literature. We found that our model is able to account for experimental data whenever it has been published (even in non trivial cases such as the *pCLV3::WUS* perturbation). Also we can with the model show that no interactions can be removed without losing connection to experiments. All this is in our opinion an important investigation of the proposed network, becoming possible with a computational approach. Finally, the model offers a host of predictions that can participate in directing future experimental work. We have extended the model description to add more arguments for the selected network interactions (page 11,12) and this is also mentioned in the discussion.

8. Small comments on the model: it is not clear to me which assumptions are made so KAN1 is not expressed in the CLV3 domain.

We thank the reviewer particular for this comment, since it is one example where a model is useful to understand if few network interaction are sufficient to explain the three dimensional experimental patterns. No assumptions *per se* are made to prevent the expression of *KANI* in the cells of the CLV3 domain. The model is designed (from data) so that WUS activates *CLV3* and represses *KANI* (which is the only difference between the transcriptional regulation of these genes), and it might not be obvious that a single WUS gradient is sufficient to explain the spatial (wild type) expression patterns as well as the dynamics from all perturbations. In all our model solutions (optimized parameter values), the patterns are obtained when lower

levels of WUS are sufficient to repress *KANI* than to activate *CLV3*, allowing for separation of the *CLV3* region from the PZ. Different binding affinities of WUS on the promoters of *KANI* and *CLV3* could possibly explain this behavior. We have now added a comment on this in the discussion (page 19).

9. The authors state: " Thus far, our analysis suggests that stem cell maintenance may be a systems property that arises as a result of collective repression of differentiation promoting transcription factors...". However, the model is quite deterministic about the initial position of all the components and the signals that trigger them, so it is difficult to see it as a "systems property" rather than as an effect of their deterministic model.

We thank the reviewer for this comment. The sentence was referring to the experimental findings up to this point, and we agree that it was not very well formulated. We have reformulated it to “Thus far, our analysis suggests that stem cell maintenance may partly be a result of collective repression of differentiation promoting transcription factors.” (page 11).

2nd Editorial Decision

10 January 2013

Thank you again for submitting your revised work to Molecular Systems Biology. We have now heard back from the two referees who accepted to evaluate the study. The referees are now supportive and we will be able to accept your paper for publication pending the following minor points:

- Reviewer #1 asks to add a scale bar to Fig 5.
- The microarray data should be deposited in a public database. Please add a 'Data availability' subsection to Materials & Method and list the respective accession numbers.
- We could not open the files for datasets 1, 4, 5, 6
- We appreciate that you link to <http://dev.thep.lu.se/organism/>. We could however not find on this page the links to download the simulation software and the specific model developed in this study. For long-terms archival purposes, we would thus kindly ask you to supply all the scripts and source code required to run and reproduce the computational analysis presented in this paper. You can upload this as a zip compressed folder. Please make sure that you include a README file at the top level.

Referee reports:

Reviewer #2 (Remarks to the Author):

The authors have addressed the majority of the issues raised by the reviewers with additional data and discussion. The additional ChIP-qPCR experiments and analysis of TAAT binding elements has bolstered the argument for WUS binding. There is one minor point that still needs addressed; figure 5 is still missing a scale bar. The additional data and clarifications have improved the manuscript and I am happy to support it for publication.

Reviewer #3 (Remarks to the Author):

I am largely satisfied with the comments that authors made in response to my concerns and suggestions. Therefore, I am happy to conclude that the manuscript in its current form is suitable for publication in MSB

2nd Revision - authors' response

30 January 2013

Thank you for the reviews and comments. We have now addressed all minor points for completing our submission.

We have:

- Updated Figure 5 and 6 with scale bars, and updated the figure captions.
- Added a section Data availability and added a link (with accession number) for the microarray data.

- Replaced data sets (Supplemental Tabs) 1, 4, 5, 6 with files stored in .xlsx. Please notify us if the files are still not readable for you (we have been able to open the new (and old) files using Microsoft Excel on a Mac).
- Uploaded MSB_CWK.zip including our simulation tool and scripts for our computational work. The archive includes a README file at the top level explaining how to compile and use the software. The description on how to access this as supplemental material is added to the new Data availability section in the manuscript.